# SHANKS: SIMULTANEOUS HEARING AND THINKING FOR SPOKEN LANGUAGE MODELS

## ABSTRACT

Current large language models (LLMs) and spoken language models (SLMs) begin thinking and taking action only *after* the user has finished their turn. This can create a high latency for waiting until the model ends the thinking process. Consequently, thinking *after* receiving the full input is not suitable for speech-to-speech interaction, where real-time and low-latency interaction is important. We address the above issue by drawing inspiration from the fact that humans can naturally *"think while listening"*. In this paper, we propose SHANKS, a general inference framework that enables SLMs to generate unspoken chain-of-thought reasoning when listening to the user input. SHANKS streams the input speech in fixed-duration chunks and, as soon as an input chunk is received, generates an unspoken reasoning based on all previous speech and reasoning; in the meantime, the user is still speaking. SHANKS uses unspoken reasoning to perform intermediate calculations, make API calls to complete the task, and determine whether to interrupt the user. We demonstrate that SHANKS enhances the real-time user-SLM interaction in two scenarios: (1) When the user is presenting their solution to a math problem, SHANKS can listen to and reason over the user's speech and make interruption when the user makes a mistake. SHANKS interrupts the user 37.1% more accurately compared with a baseline that interrupts the user without thinking. (2) In a task-oriented dialogue setting, where the user's request needs to be completed by calling hotel and flight booking APIs, SHANKS can complete 63.2% of the API calls before the user even ends their turn. In summary, SHANKS provides a promising perspective to thinking when the user is still speaking time to improve user-LLM interaction.

## 1 INTRODUCTION

In recent years, the *thinking* process has been used to improve Large Language Models (LLMs), where the LLM first generates a *hidden* chain-of-thought (CoT) reasoning (Wei et al., 2022; Kojima et al., 2022) invisible to the users, and then generates the final output response (OpenAI, 2024b; Guo et al., 2025). This thinking process improves LLMs on reasoning-intensive tasks, including mathematics (Lightman et al., 2024), coding (Chen et al., 2021), and questions that involve significant domain knowledge (Rein et al., 2024). However, current reasoning LLMs only start to think *after* receiving the complete user input, which is reasonable for turn-based interactions, i.e., the model processes the user's message after it is fully composed and sent.

In contrast, human behavior in *spoken* communication is different. Humans naturally think *while* listening, far before the speaker finishes their turn. Thinking during listening offers two key advantages: (1) It enables timely and well-founded reactions, including interruption, even before the speaker concludes. (2) It reduces response latency by allowing answer preparation to begin before the speaker finishes speaking. Motivated by these observations, we propose a method to enable spoken language models (SLMs) to think while listening to input speech.

In this paper, we introduce SHANKS: **S**imultaneous **H**earing **a**nd Thi**nk**ing with Chunked Input **S**peech. SHANKS is a general inference framework for SLMs to achieve thinking while listening, which can be obtained by fine-tuning any off-the-shelf SLMs. At inference time, SHANKS processes the user input in a fixed-size chunk. Once a chunk of speech input is received, SHANKS generates a chunk of thinking tokens based on all previous input speech chunks and previous thinking chunks.

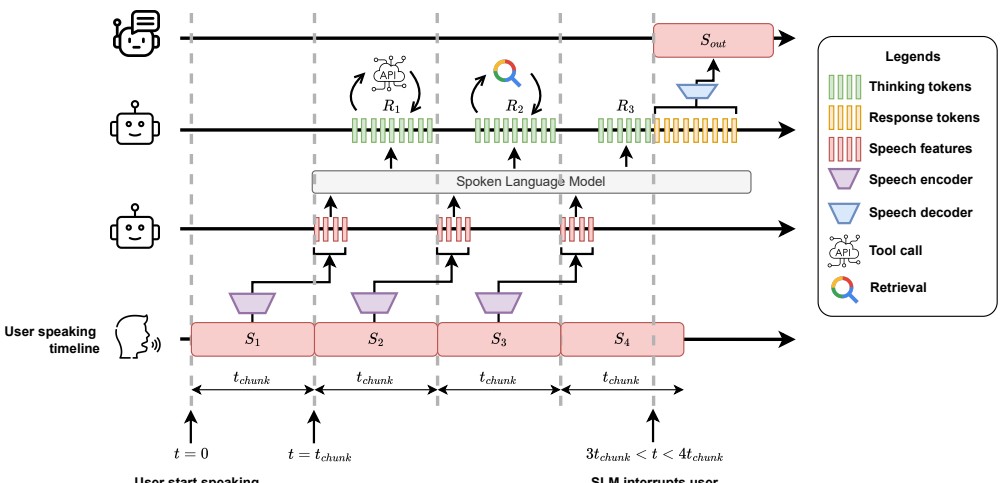

Figure 1: The timing diagram of SHANKS. As the user speaks, their speech is segmented into chunks for every $t_{chunk}$ seconds and streamed to the SLM. After receiving a speaking chunk, SHANKS generates the thinking tokens, which might include calling external tools or performing retrieval. When the user is speaking the $i$-th speech chunk $S_i$, SHANKS generates the $(i-1)$-th thinking chunk $R_{i-1}$, achieving thinking while listening. When the next speech chunk $S_{i+1}$ is fully spoken by the user, the SLM stops the reasoning for $R_{i-1}$, adds the newest speech chunk to its context, and begin the $i$-th thinking chunk. At any point, the SLM can optionally interrupt the user during their speech.

SHANKS alternates between receiving the input speech chunk and generating an unspoken thinking chunk until the user's speech finishes, and then the model generates the spoken response. When the user is still emitting a speech chunk, SHANKS uses the same time to generate a chunk of thinking, achieving the ***thinking while listening***. During the thinking process, SHANKS can decide to interrupt the user or make tool calls. The inference workflow of SHANKS is depicted in Figure 1. To the best of our knowledge, we are the first to explore generating unspoken CoT reasoning when the user is still speaking.

We use two scenarios to show how SHANKS can improve real-time user-SLM interaction and reduce latency. First, we study a scenario where the user first describes a math question and then describes their step-by-step solution. SHANKS can listen to the user's problem-solving process and perform internal thinking in the meantime to interrupt the user when the user makes a mistake in their solution. This scenario has great potential in educational use cases, where the SLM serves as a tutor to guide the student. Compared to a baseline that makes an interruption without thinking, SHANKS interrupts 71% more when the user makes a mistake, while the interruption made by SHANKS is 37.1% more valid.

Next, we focus on a task-oriented dialogue setting, where the user requests the SLMs to search for flights and hotels to fulfill the user's travel plan, and the SLM must make API calls (e.g., searching flights or booking hotels) to complete the task. SHANKS enables the model to successfully complete 63% of API calls while the user is still speaking; without SHANKS, these API calls can only be made after the user's speech ends. Hence, SHANKS can reduce the latency that the user needs to wait. We show an example of how SHANKS performs API calls during user speech in Figure 3 in the Appendix.

## 2 BACKGROUND

Thinking LLMs are trained to perform a deliberate thinking process before emitting the final text responses (Guo et al., 2025; OpenAI, 2024b; Wu et al., 2024). The thinking process is enclosed in some special tokens, e.g., `<think>` and `</think>`. The thinking process, which represents the LLM's intermediate steps to solve the user query, is internal to the LLM and hidden from the users.

Different from the rapid development of thinking LLMs, currently, almost all SLMs (speech-in-speech-out models) cannot generate a thinking process before speaking out the final response (Zeng

et al., 2024; Huang et al., 2025; Li et al., 2025; Ding et al., 2025; Wu et al., 2025). Note that SLMs can generate speech responses and are different from audio-aware language models that can only output texts (Xie et al., 2025; Chu et al., 2024). A notable exception is a concurrent work from Chiang et al. (2025). In Chiang et al. (2025), they let the SLM think when the SLM is speaking. The thinking process in Chiang et al. (2025) still happens after the user's turn ends, which is different from our thinking-while-listening.

In our paper, we will use Qwen-2.5-Omni (Qwen-omni for short) (Xu et al., 2025), one of the best open-sourced SLMs, in our experiment, and we introduce the necessary background. Qwen-omni is a *thinker-talker SLM*. The thinker takes speech representation extracted by a speech encoder (Chu et al., 2024) as the input and generates **text tokens**. The talker functions like a text-to-speech (TTS) model, taking the hidden representation from the thinker as the input and generating the output speech.

Qwen-omni is not capable of performing *unspoken* thinking – every token (and its corresponding hidden representation) generated by the thinker will be sent to the talker model and synthesized into speech. As the main goal of the paper is to make SLMs think silently while listening, where the thinking token should not be spoken out, we will need to teach Qwen-omni to generate unspoken thinking tokens. To do so, we will fine-tune the **thinker** model on a speech-in-**text**-out dataset to teach the model to generate the thinking process.

Here, we unify the terms that will be used throughout the paper. *Thinking* tokens/chunks refer to the output from the thinker enclosed within `<think>` and `</think>`. *Response* tokens/chunks refer to the thinker output (text) tokens outside of the `<think>`/`</think>` markers. Thinking tokens and response tokens are both text tokens, and they differ in whether they will be spoken out by the talker model.

Since we only want the Qwen-omni to speak out the response tokens, we only pass the response tokens to the talker. After fine-tuning the thinker, it will be necessary to adapt the talker so it can recognize the output from the fine-tuned thinker. Fine-tuning the talker model is quite straightforward and not the focus of this paper, i.e., thinking while listening, and we do not discuss how to adapt the talker here.

To make Qwen-omni think while listening, the main burden lies in how to make the *thinker* model generate thinking tokens when the user is speaking. As a result, we mainly focus on how **thinker** should be trained and how it should be used during inference. In the rest of the paper, **we will use SLM to specifically refer to the thinker of Qwen-omni**, which takes user input speech and generates thinking and response tokens.

## 3 METHOD: SIMULTANEOUS HEARING AND THINKING WITH CHUNKED INPUT SPEECH

Current LLMs and SLMs only start to think after the user's input is completed. In contrast, humans can think while listening, where we reason over what we just heard, guess what the speaker might be up to, and prepare the necessary ingredients to cook up a good response. Thinking while listening allows us to react to the speaker better when the speaker is still speaking. In this section, we introduce SHANKS, a general framework to make SLMs capable of thinking while listening. Here, we only discuss the basic form of SHANKS, and we defer the more advanced usages, including interruption or tool call, to later sections.

### 3.1 INFERENCE

During inference, SHANKS requires that the user's input speech comes in a streaming manner. SHANKS processes the streaming user input speech by a fixed chunk size of $t_{\text{chunk}}$ seconds. We use $S_i$ to denote the $i$-th user input speech chunk, where $S_i$ is an audio chunk of $t_{\text{chunk}}$ seconds, except for the last chunk $S_N$, which may be shorter. When the user is still speaking, SHANKS alternately takes the user speech $S_i$ and generates the thinking chunks $R_i$ conditioning on all previous user speech and all previous thinking chunks.

Here, we walk through what happens for SHANKS during inference. The following contents are best read with Figure 1. At $t = 0$, the user begins to talk. When $t = t_{\text{chunk}}$, the user speech from 0 to

$t_{\text{chunk}}$, i.e., $S_1$, is sent to the SLM. Here, we append a special token `[EOPA]` (end of partial audio) after $S_1$ to let the SLM know that this is the end of a chunk of *partial* user speech. Based on $S_1$, the SLM generates the first thinking chunk $R_1$. A thinking chunk is enclosed in two special tokens `<think>` and `</think>`. The SLM generates $R_1$ during the interval $t = t_{\text{chunk}}$ to $t = 2t_{\text{chunk}}$, and the user is still speaking the second chunk $S_2$ at the same time. Since $t_{\text{chunk}}$ is the time for the SLM to generate its thinking, the duration of $t_{\text{chunk}}$ cannot be selected too small; otherwise, the SLM may not be able to produce meaningful thinking chunks. Unless specified, we select $t_{\text{chunk}} = 4.0s$ in our paper; a 7B model can generate around 320 tokens on a single A100 GPU in this duration.

At $t = 2t_{\text{chunk}}$, we take the freshly obtained user speech chunk $S_2$ (from $t = t_{\text{chunk}}$ to $t = 2t_{\text{chunk}}$) and pass this chunk to the SLM, and again appending the `[EOPA]` after this chunk. (Assume that the user still has not ended their turn at $t = 2t_{\text{chunk}}$.) When generating the next thinking chunk $R_2$, the SLM conditions on $S_1$, $R_1$, *and* $S_2$. The SLM will continue the process of taking user input speech chunks and generating the thinking chunks until the user ends their speech in the $N$-th chunk of speech, $S_N$. After the user ends their speech, we feed the last speech chunk $S_N$ into the SLM, while this time we append a different special token `[EOA]` (end of audio), indicating that the user's speech has ended. Based on $S_N$ and all the previous interleaved speech/thinking chunks $\{S_1, R_1, \cdots, S_{N-1}, R_{N-1}\}$, the SLM generates the thinking chunk $R_N$ and then generates a final response chunk $O$.

Since SHANKS chunks the user input using a fixed-duration chunk $t_{\text{chunk}}$, the model's thinking will lag behind the user's speech by at least $t_{\text{chunk}}$ seconds. If the user's speech is less than $t_{\text{chunk}}$, SHANKS cannot think while listening. However, since long speech can easily happen in real-world interaction, this limitation might not be a significant weakness of SHANKS.

## 3.2 TRAINING

During inference, SHANKS requires the SLM to generate thinking chunks based on all previous user input speech chunks and the model's own thinking chunks. During training, we prepare datasets to make the SLM learn this behavior. Assume that we have a complete user speech $S$, we can segment it into $N$ chunks $\{S_1, \cdots, S_N\}$ with a fixed duration $t_{\text{chunk}}$. Next, assume that we use some method to obtain the thinking chunks $\{R_1, \cdots, R_N\}$ and the output response $O$, which we will explain in later sections. We use the standard language modeling cross-entropy loss to train the SLM to predict $R_1$ given $\{S_1\}$, predict $R_2$ given $\{S_1, R_1, S_2\}$, and predict $R_N$ and $O$ given $\{S_1, R_1, \cdots, S_{N-1}\}$.

## 4 SCENARIO 1: INTERRUPTING USER TURN

After introducing the basics of SHANKS, we use two tasks to demonstrate how SHANKS can be applied. In the first scenario, we aim to use SHANKS to make SLMs able to interrupt the user based on their thinking when the user is saying something wrong. The significance of this application lies in its potential in educational use cases, where the SLMs can serve as a tutor and listen to the speaker, a student, describing how they solve a problem. The SLM can make a timely interruption to let the student know that they are making a mistake, allowing them to correct it as early as possible.

This application is highly related to the full-duplex ability of spoken language models (Lin et al., 2025). While most prior works on full-duplex SLMs focus on user interrupting SLMs, we focus on the reverse scenario: SLM interrupting users. As an important note, we do not advocate that it is good to have a model that interrupts the user. Some users might find it annoying and unpleasant when interrupted by an SLM. Here, we are merely introducing how to make the model capable of interrupting the user from a modeling perspective, while the model deployers and the users can choose whether to enable this interruption behavior.

### 4.1 TASK DESCRIPTION

We explain the precise task we are evaluating. In this task, the user describes a math problem and then solve the problem. The user's solution does not simply state the answer; the user describes a step-by-step problem-solving process, which might be correct or wrong. The SLM needs to interrupt the user when the user is making a mistake, and not to interrupt the user when the solution is correct.

As this is a novel task and there is no available data, we built the evaluation data ourselves. First, we construct the user speech, which should include a math question and a step-by-step solution. We source the math questions from the testing data of GSM8K (Cobbe et al., 2021), a grade-school math word problem dataset commonly used for evaluating mathematical reasoning ability (Wei et al., 2022; Kojima et al., 2022; Wang et al., 2023). Next, we use two LLMs, Llama-2-7B (Touvron et al., 2023) and Llama3.1-8B (Grattafiori et al., 2024), to generate step-by-step answers for those questions, and use GPT-4o to determine if the answer generated by the two models matches the ground truth answer in the dataset. We select these two models since they can generate CoT reasoning to solve the math problem, and their performance on GSM8K is very different: Llama-2-7B is a weaker model and prone to generating wrong solutions, while Llama-3.1-8B is a stronger model, which can generate more accurate solutions.

After we have the texts for the step-by-step solution, we convert them into speech. We use GPT-4o to rewrite the answers generated by the two Llama models to make the solution more colloquial. Next, we concatenate the original question, the colloquial step-by-step answer, and prepend a prefix *"I want to solve the following question."* to form the transcription of a testing instance. We use GPT-4o-mini-TTS (OpenAI, 2024a) to synthesize the speech.

The final testing dataset includes 1280 instances with correct solutions and 1140 with incorrect solutions. We call the former the *"correct subset"* and the latter the *"incorrect subset.* The average duration of the user speech is around 49.25 seconds.

## 4.2 Training Data for Interruption

To teach the model to think while listening and determine whether to interrupt, the training data in this task include two types of instances: (1) The user provides a correct step-by-step solution to the question, and the model does not interrupt the user during the user's speech. After the user finishes the speech, the output response acknowledges the correctness of the answer. (2) The user's turn unfolds an erroneous problem-solving process, and the model interrupts the user when the user makes the first mistake and clearly explains what is wrong.

To construct such a training dataset, we use the math questions in Tulu3-Persona-Math-Grade (Lambert et al., 2024) to construct the user speech $S$ following the previously described procedure, and then segment the speech by a fixed duration $t_{\text{chunk}} = 4$ seconds to obtain $\{S_1, \cdots, S_N\}$.

We use GPT-4o to generate the thinking chunk $R_i$. When generating the $i$-th thinking chunk $R_i$, the input to GPT-4o includes the transcriptions of all previous user speech chunks $\{S_1, \cdots, S_i\}$ and all previous thinking chunks $\{R_1, \cdots, R_{i-1}\}$. GPT-4o is required to do the following in the thinking chunk: (1) Track the information already known and calculate intermediate variables when they are available. (2) Identify if any errors are present in the user's current transcription. If there is an error, GPT-4o should generate a [INTERRUPT] token at the end of the thinking chunk, indicating that the user should be interrupted. We give GPT-4o four in-context examples to allow GPT-4o to understand the task.

After generating the thinking chunks, we generate the final output response $O$. For the user speeches with an error-free solution, the output response simply needs to let the user know that their solution is correct. We prompt GPT-4o to generate the final response based on the full user speech and all previous reasoning. Now, we can form a training data sequence by interleaving $S_i$ and $R_i$ and then appending $O$ in the end.

For those user speeches with a wrong solution, the output response will be an interruption to the user's speech. Assume that based on our previous process for generating the $R_i$'s, GPT-4o decides to interrupt the user after the user speech chunk $S_k$, i.e., the thinking chunk $R_k$ includes the interruption token [INTERRUPT]. To generate a response for interruption, we give GPT-4o the user's speech up to the $k$-th user speech chunk, all the previous thinking, and ask GPT-4o to generate a response $O$ to interrupt the user. The interruption should be precise on what error is made by the user and how to correct it. After this process, we can interleave $S_1$ to $S_k$ with $R_1$ to $R_k$ and append $O$ in the end to form a training sequence. A figurative illustration of this training instance is shown in Figure 4(b) in the Appendix. Note that in the last thinking chunk $R_k$, there will be a special token [INTERRUPT], indicating that the user is going to interrupt the user.

## 4.3 EVALUATION

During inference, we stream the user speech to the SLM by a fixed chunk size $t_{\text{chunk}}$, and follow the inference procedure elaborated in Section 3.1. If the SLM generates the special token [INTERRUPT] in a thinking chunk $R_k$ and outputs a response chunk $O$ (when the user is emitting speech chunk $S_{k+1}$), we convert the response token into speech to interrupt the user.

We evaluate a model on the testing dataset constructed in Section 4.1. We use the following metrics, separately reported for the correct and wrong subsets:

**(1) Interrupt ratio**: The ratio of total interrupted instances among the total instances. A good model should have a low interrupt ratio on the correct subset and a high interrupt ratio on the wrong subset.

**(2) Valid interrupt ratio**: The number of *valid* interruptions among the total interruption. To judge whether an interruption is valid or not, we use LLM-as-a-judge (Chiang & Lee, 2023; Zheng et al., 2023). We give the judge LLM the user input until the time of interruption[1] and output response $O$ from the model, and ask the LLM judge to determine if the model's interruption response $O$ correctly interrupts the user when there is unclear or mistakes in the user speech.

**(3) Interruption latency**: The time of the model interruption compared to *the time when the first error happens in the user's speech*, denoted as $t_{\text{error}}$. For samples in the incorrect subset, use GPT-4o to determine $t_{\text{error}}$. The details on this procedure are included in Appendix C.1. For the correct subset, there are no errors in the user's speech, and the $t_{\text{error}}$ is defined as the duration of the audio. Assume that the model interrupts the user at $t_{\text{interrupt}}$, then the interruption latency is calculated as $t_{\text{interrupt}} - t_{\text{error}}$. $t_{\text{interrupt}}$ is the time when the first response token is generated. In fact, a more accurate $t_{\text{interrupt}}$ should consider the time the talker synthesize the audio from the response tokens. However, since the time for synthesizing the audio is fixed across all the model we consider here, we omit this latency when calculating $t_{\text{interrupt}}$. A negative latency means that the model interrupts the user when the user has not made an explicit error.

## 4.4 EXPERIMENTS SETUP

We fine-tune the thinker model in Qwen-omni on training data constructed in Section 4.2 with 5K samples. We refer to this model as SHANKS-E2E to draw a distinction with another cascade version to be introduced later. Other training details and hyperparameters are included in Appendix B.1. Since there are no other models that can interrupt the user[2], we fine-tune two baselines and compare them with SHANKS-E2E.

(1) **No-thinking**: We fine-tune Qwen-omni to predict whether it should interrupt the user without any thinking. The model is trained to predict a special token, [NO_INTERRUPT] or [INTERRUPT], to indicate whether the model should interrupt the user, given chunked user input speech. This can be think as SHANKS while the thinking chunks only contain a [NO_INTERRUPT] or [INTERRUPT] special token.

(2) **SHANKS-Cascade**: We set up a cascade version of SHANKS. Precisely, we cascade an ASR with a stronger text-only LLM, Qwen-2.5-7B-Instruct (Qwen et al., 2025), to make the LLM generate thinking chunks while reading the partial transcription. Qwen-2.5-7B-Instruct and Qwen-omni are fine-tuned from the same base model, while Qwen-2.5-7B-Instruct are fine-tuned on a much larger reasoning dataset. We fine-tune Qwen-2.5-7B-Instruct on a similar text-input dataset. This baseline allows us to know what the performance of SHANKS can be if we use a model with better reasoning ability as the backbone.

## 4.5 EXPERIMENT RESULTS

The results are presented in Table 1. We have the following observations.

**SHANKS is more likely to interrupt on the wrong subset.** Comparing the interruption ratio of SHANKS on the correct and wrong subsets, the interruption ratio is 54.2% higher on the wrong subset.

---

[1]If the interruption happens in $R_i$, the user is currently speaking the $(i + 1)$-th speech chunk $S_{i+1}$, as the thinking $R_i$ happens simultaneously when the user says $S_{i+1}$. Consequently, we also feed the transcription of $S_{i+1}$ into the judge model when determining whether the interruption is valid.

[2]We found that closed-source models like GPT-4o cannot interrupt the user when the user is still talking, not to mention open-source SLMs.

| Subset | Correct Subset (1280) | | | Wrong Subset (1140) | | |
|---|---|---|---|---|---|---|
| Metrics | Interrupt ratio (%) ($\downarrow$) | Valid interrupt ratio (%) ($\uparrow$) | Interrupt latency (s) | Interrupt ratio (%) ($\uparrow$) | Valid interrupt ratio (%) ($\uparrow$) | Interruption latency (s) |
| SHANKS-E2E | 30.6% | 25.7% | -4.24 | 84.8% | 63.9% | 5.08 |
| No-thinking | 1.4% | 16.7% | -5.68 | 13.8% | 26.8% | 6.46 |
| SHANKS-Cascade | 24.9% | 40.3% | -3.0 | 86.1% | 78.3% | 6.90 |
| *Ablations for* SHANKS-*E2E* | | | | | | |
| $t_{\text{chunk}} = 3.0$ | 41.1% | 21.4% | -3.33 | 88.7% | 60.3% | 1.56 |
| $t_{\text{chunk}} = 5.0$ | 26.9% | 36.9% | -11.38 | 83.1% | 66.2% | 8.19 |

Table 1: Results for interrupting the user. We report the interruption ratio and valid interruption ratio in percentage, and the interruption latency in seconds. $t_{\text{chunk}} = 4.0$ in the top three rows.

This shows that SHANKS is indeed capable of capturing the errors in the user's speech and interrupt appropriately. Based on the valid interruption ratio for the wrong subset, 2 out of 3 interruptions made by SHANKS are valid. Interesting, on the correct subset, the valid interruption ratio is non-zero. By looking into the instances in the correct subset, we find that even if their final answers are correct, sometimes their intermediate reasoning may be odd or ambiguous, and the model will interrupt and ask for clarification. Prior works also reported that even if the final answer of the model is correct, the CoT reasoning may be wrong (Golovneva et al., 2023). In this case, the LLM judge treats this kind of interruption as valid.

**SHANKS interruption latency shows that the model mostly interrupts after the error occurs.** On the wrong subset, the interruption latency is 5.08 seconds on average. In Figure 5 in the Appendix, we further plot the distribution of the interruption latency. We find that the interruption latency on the wrong dataset is left-skewed, where more samples fall on the right proportion of the distribution and have a positive interruption latency. This indicates that most interruption happens later than the first error.

**Interruption without thinking leads to much poorer performance.** The performance of the no-thinking baseline is much worse than SHANKS, which performs reasoning before interrupting. The no-thinking baseline has a much lower interruption ratio on the wrong subset, and the valid interrupt proportion is also much lower than SHANKS. This shows that thinking before interruption is important, justifying the design of SHANKS.

**Cascade version of SHANKS with stronger LLM leads to the best performance.** When using Qwen-2.5-7B as the backbone model for SHANKS, the performance can be even better. The interruption ratio on the correct subset is lower, and the valid interruption ratio on the wrong subset also grows higher. This shows that the interruption ability of SHANKS is mostly related to the reasoning ability of the backbone model, and using a stronger reasoning LLM can improve the performance.

**Varying $t_{\text{chunk}}$ at inference time does not significantly affect the performance.** When constructing the training data, we fix $t_{\text{chunk}} = 4$ seconds. Here, we ask whether we can vary $t_{\text{chunk}}$ at inference time. Since the thinking of SHANKS always lags behind the latest user speech by $t$ seconds, changing $t_{\text{chunk}}$ can affect how soon the SLM can hear the latest user speech and affect the response latency. As an ablation, we change $t_{\text{chunk}}$ to 3 and 5 during inference without retraining the model. The results are shown in the bottom two rows in Table 1. On the wrong subset, we do not find the interrupt ratio and valid interrupt ratio to change significantly compared with $t_{\text{chunk}} = 4$. Interestingly, we find that the interrupt latency on the wrong subset for $t_{\text{chunk}} = 3$ is the smallest, while the $t_{\text{chunk}} = 5$ has the largest interrupt latency.

## 5 SCENARIO 2: API CALL WHEN LISTENING

In the second scenario, we focus on a task-oriented dialogue setting, where a user describes their travel plans, and a customer service agent needs to make relevant API calls to complete the request and respond to the user. An example input is: "*Help me check the details of the cheapest flight*

*from Hangzhou to Seoul on December 10, 2024, and the car rental information near Seoul airport.*"
To complete this request, the service agent need to call APIs to search for the flight and car rental
information. Current LLMs will complete this task by waiting until the full user input is completed,
and then start to call APIs to search for relevant information, and then formulate a final response
based on API call responses.

However, one can observe that when the user is halfway through their speech, the search flight API
can already be called since the destination and date is already clear. This is where SHANKS can be
useful: processing partial user input and perform early actions. In this task, we will give the model
a user request the requires to make API calls, and we will evaluate the model's ability to correctly
make the API calls and how fast those API calls can be made.

## 5.1 TASK DESCRIPTION

To evaluate SLM's ability to make API calls based on the user input, we adopt Complex-
FuncBench (Zhong et al., 2025). ComplexFuncBench is a complex tool-calling benchmark. A
tested model will be given a user query specifying some requirements for a travel plan, including
flight searching, hotel reservation, etc, and a list of potential APIs that are required to complete the
task, including APIs for searching hotels or flights. The goal of the model is to make relevant API
calls and provide the information to the user. ComplexFuncBench provides the ground truth API call
(the API calls that must be made to complete the task) for each instance and the API responses for
those ground truth API calls, which can be used to evaluate whether the model's API call is correct.
To adapt ComplexFuncBench in our use case, we use GPT-4o-mini-TTS to synthesize the user speech
instruction from the text instructions from ComplexFuncBench.

## 5.2 TRAINING DATA FOR API CALL

To train SHANKS to perform API calls when listening, we need to teach the model to make API calls
based on user input speech chunk $S_i$ when the information for the API call is complete. We split half
of the instances in ComplexFuncBench to construct the training data and the other half as the testing
data.[3] The speech chunks $S_i$ in the training data can simply be obtained from chunking the audio of
the complete speech.

In this task, the thinking chunk $R_i$ is simply the API calls and responses based on the previous speech
and thinking chunks $\{S_1, R_1, \cdots, S_i\}$. Since ComplexFuncBench already provides the ground truth
API calls to fulfill the task, we only need to determine which API call can be made after a speech
chunk $S_i$. To determine which API calls can be made in $R_i$, we give GPT-4o the transcription of the
speech chunks from $S_1$ to $S_i$, and ask GPT-4o whether the user's speech already provide sufficient
information about what API call. If GPT-4o thinks the API call can be made, the API call and
response are added to $R_i$. If no API calls can be made in $R_i$, we put a template message that says
there are no additional tool calls that can be made.

The final response $O$ is also generated by GPT-4o by prompting it to generate a final response based
on the instructions and all the ground truth API calls and responses. During training (and also
inference), the descriptions of the API calls necessary to complete the user's request will be included
in the system prompt.

## 5.3 EVALUATION

During inference, we feed the user speech to a tested model, and let the model make API calls. Given
an API call made by the model, we use GPT-4o as a judge to determine if the API call matches one
of the ground truth API calls, and return the response of the ground truth API call if a match is found.
Using GPT-4o to match the API call made by the model against the ground truth follows one of the
evaluation protocols in the original ComplexFuncBench.

We evaluate the performance of the model on 500 testing instances using two metrics: (1) **Call
accuracy**: The number of ground truth API calls that are successfully made by the model, divided by

---

[3]ComplexFuncBench is originally designed as an evaluation dataset. Here, we train the model directly on this
dataset since Qwen-omni was not trained to perform tool call. Since our training data has the same distribution
as the testing data, our results should not be compared with other models that are not trained on this dataset.

the total number of ground truth API calls in the dataset. We also calculate the *early call accuracy*, defined as the ground truth API calls that are successfully made by the model *when the user is still speaking*, divided by the total number of ground truth API calls. Similarly, we calculate the *late call accuracy*, where the dividend is the ground truth API calls that are successfully made *after the user finishes speaking*. This helps us understand how well the model leverages the time when the user is still speaking. (2) **Task success rate**: The percentage of task that are successfully completed. If all the ground truth API calls in an instance are made, the task is considered successful.

In this task, we only focus on whether the API calls made during the thinking process are correct, and we do not assess the final response of the model.

## 5.4  EXPERIMENT SETTING

We compare two models, both fine-tuned from Qwen-omni. (1) SHANKS: fine-tuned from the training data in Section 5.2. SHANKS can perform API calls and get the responses when the user is still speaking. (2) "*Call-after-listen*": In this baseline, the model is trained to take the full user speech and iteratively makes the API calls, takes the responses, and makes new API calls until the task is completed.

| Methods | Call Accuracy (%) | | | Success rate (%) |
|---|---|---|---|---|
| | Early | Late | Total | |
| SHANKS | 58.3% | 15.0% | 73.3% | 37.0% |
| Call-after-listen | 0 | 88.1% | 88.1% | 62.4% |
| SHANKS + call-after-listen | 63.2% | 25.2% | 88.4% | 72.2% |

Table 2: Results for API calls. The early and late do not sum to 100 since the denominator is the total ground truth API calls that should be called, and some ground truth API calls may not be successfully called.

## 5.5  EXPERIMENT RESULTS

The experiment results are presented in Table 2. We find that SHANKS can successfully make 58.3% of the API calls when the user is still speaking. Compared with the call-after-listen baseline, SHANKS can greatly reduce the time the user needs to wait. This is because call-after-listen needs to wait until the user has finished to start to make API calls, while many calls can readily be made when the user is speaking.

However, the success rate of SHANKS is lower than call-after-listen. We find that this is because SHANKS rarely retries the failed API call attempts, while the call-after-listen baseline tends to iteratively retry failed API calls.

To solve the above issue, a simple method is to use SHANKS when the user is still speaking and back off to call-after-listening when the user speech ends. Precisely, when the user is still speaking, we use SHANKS to call APIs while listening, and only keep the success API calls and their responses. When the user finishes their speech, we switch to the call-after-listening mode, where the input to the SLM is the complete user speech and the success API calls and responses made by SHANKS, and the model continues to make the remaining API calls. Since some API calls have already been made by SHANKS when the user speaks, this combined method enjoys the thinking-while-listening advantage of SHANKS.

In the last row in Table 2, we show the result of combining SHANKS with call-after-listening. This combined method has a high number of early call accuracy while also having a high task success rate. Compared to call-after-listen, where all the API calls need to be made after the user speech ends, the combined method can successfully call 63.2% APIs when the user is still speaking, while the remaining 25.2% API calls will be called after the user's speech finishes. This means that the combined method can reduce the user wait time while maintaining the performance.

## 6  CONCLUSION

In this paper, we introduce SHANKS, a framework that enables SLMs to think while listening. SHANKS achieves thinking-while-listening by chunking the user input speech and progressively reasoning over the user input based on all previous thinking processes. When the user is speaking, the SLM is generating thinking chunks for all previous input speech, achieving thinking while listening. On two scenarios, we show that SHANKS enables more accurate interruption behavior and can make API calls when the user is still speaking. We believe SHANKS open new possibility of SLMs.

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

## A  Encoding the User Speech

In the main content of the paper, we say that we chunk the user input audio into fixed-size chunks of $t_{\text{chunk}}$ seconds. In fact, what we do is perform chunking at the level of feature representation instead of the level of audio waveform. Precisely, when encoding the $i \geq 2$ speech chunks $S_i$, we feed the full speech through the audio encoder, and only take the speech representation for the corresponding speech chunk. If we directly chunk the audio waveform and encode each audio chunk independently, the representation of later audio chunks will not be able to depend on the earlier audio chunks, which can potentially lead to performance degradation.

## B  Details in Training

We fine-tune the models using the Llamafactory (Zheng et al., 2024) toolkit. One thing we omit in the main content is that when generating the training data using GPT-4o, we do not feed the audio of the user speech into GPT-4o. Instead, we feed the transcription of the speech chunks. This is because using the speech chunk will increase the cost and time to call the API. To obtain the transcription of each chunk, we use Whisper-large to obtain the transcription and timestamp for each word in the user speech, and then segment the transcriptions into chunks based on the timestamp.

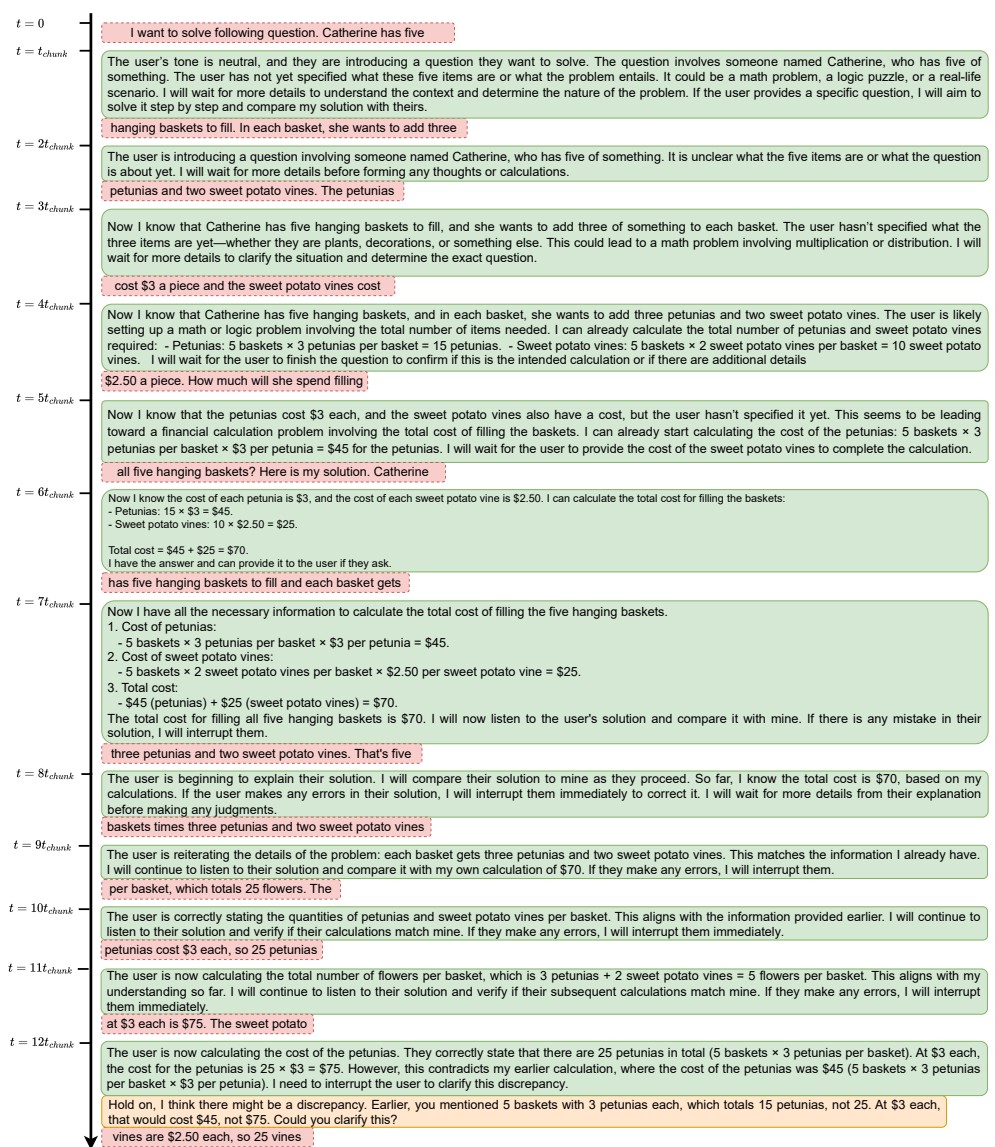

Figure 2: Examples of the evaluation samples in the interruption scenario, including the thinking and response from SHANKS. For each time slot from $nt_{\mathrm{chunk}}$ to $(n+1)t_{\mathrm{chunk}}$, the chunks in green (SLM thinking chunks), blue (API call responses), and orange (output response) happen *sequentially*, while the user speech chunk (in red) happens concurrent to other blocks in the same time slot.

## B.1 FINE-TUNING FOR INTERRUPTION

To prepare the training data, we randomly sample 5K samples from Tulu-3-SFT-Math-Grade (Lambert et al., 2024), which can be loaded from Huggingface datasets (Lhoest et al., 2021). We follow the procedure detailed in Section 4.2 to construct the training data. We additionally filter out audios that are longer than 80 seconds, so the final training dataset is slightly less than 5K.

We fine-tune the thinker on the training data for two epochs on 8 A100 GPUs. The effective batch size is 64. We set the learning rate to $1.0e-4$ with cosine learning rate scheduling and a 0.1 warm-up ratio.

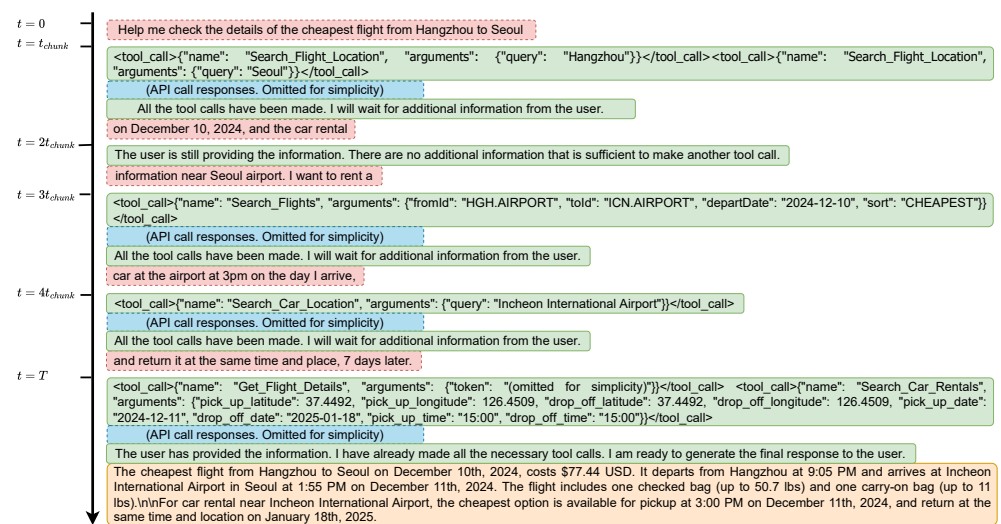

Figure 3: Examples of the evaluation samples in the API call scenario, including the thinking and response from SHANKS. For each time slot from $nt_{\text{chunk}}$ to $(n+1)t_{\text{chunk}}$, the chunks in green (SLM thinking chunks), blue (API call responses), and orange (output response) happen *sequentially*, while the user speech chunk (in red) happens concurrent to other blocks in the same time slot. The $t = T$ means the time when the user's speech terminates.

A significant part of the training data is generated by prompting GPT-4o. We include the prompts here. We list the prompt to generate the reasoning chunks in Table 3 and 4, the prompt to generate the interruption in Table 5, and the prompt to generate the response without interruption in Table 6.

### B.2 FINE-TUNING FOR API CALLS

We use the procedure detailed in Section 5.2 to construct the training data. The training data consists of 500 samples. The prompt used to determine when an API call can be made is shown in Table 8. The prompt used to generate the final response is shown in Table 9.

We fine-tune the model using LoRA (Hu et al., 2022), as the sequence length for this dataset is very large and full fine-tuning will result in out-of-memory. We also fine-tune the LM head and the token embedding of the talker model; otherwise, the model will not be able to recognize and generate special tokens. As the training dataset is smaller, we fine-tune the model for 10 epochs, while other training hyperparameters follow those in Appendix B.1.

## C DETAILS IN EVALUATION

### C.1 EVALUATION DETAILS FOR INTERRUPTION

To deteermine the time of interruption, we apply the following procedure. We use Whisper-large (Radford et al., 2022) to obtain the timestamp of each word in the user speech, and we use GPT-4o to determine when the first error in the user speech occurs by giving GPT-4o the question, the ground truth answer, the transcription of the user speech, and the word-timestamp alignment. The prompt used to determine the first error time $t_{\text{error}}$ is shown in Table 7.

The prompt used to determine whether an interruption is valid is shown in Table 10. Note that if an interruption occurs in $R_i$, the user is saying $S_{i+1}$ at the current point, and the user speech used for judgement will include the the latest speech chunk $S_{i+1}$. When calculating the latency, we do not consider the time for the talker to synthesize the audio. Since this latency due to the talker is fixed for all the models we compare, excluding the latency will not affect our comparisons in Table 1.

```
# Generate Internal Thinking While Listening

## Task Introduction
Humans are capable of thinking while listening to others speak.  Based on the partial information received, we parse important
details, clarify ambiguities, recall relevant facts, and compute intermediate variables.  Your task is to simulate this
process.  You will be the previous chunks of user's speech in text, and you will also see your previous inner thinking when
listening to those chunks.  Your job is to generate the next internal thinking as if you had listened up to the newest chunk.

When generating internal thinking spans, follow these guidelines:
1.  The inner thinking span should be fewer than 400 words.
2.  Your internal thinking should reflect the user's emotion, intent, and what you already know from the user.  If any
relevant information can be recalled or intermediate variables can be calculated based on current information, include them
in your inner thinking.
3.  The inner thinking should read more like full, coherent sentences rather than shorthand notes.  Using short notes will be
very hard to understand and possibly making logical errors.
4.  If the user's query involves a question, you **must generate your own step-by-step answer in the internal thinking before
the user finishes speaking**.
5.  Later internal thinking spans must not repeat information already covered in earlier ones.  However, if later
transcription spans update or contradict earlier information, explicitly point that out and correct it.  You may start with
phrases like \Wait, the user previously..., but now...".
6.  Always think independently in your internal thinking.  When the user is providing there solution, you should have you own
solution and then compare your own solution with the user's solution.  If you identify any error, you should interrupt the
user immediately.  Indicate the interruption by ending your internal thinking with the special token [INTERRUPT].

---

## Samples

### Example 1:
User (partial) input transcription 1
Betty is saving money for a new wallet which

Prior Inner Thinking 1
The user's tone is neutral.  The user describes a situation where someone named Betty is saving money for a new wallet.  The
user hasn't finished yet.  Perhaps they want me to give advice on how to save money.

User (partial) input transcription 2
costs $100.  Betty has only half of the money she needs.

Prior Inner Thinking 2
Now the user gives more information.  We know the wallet Betty wants to buy costs $100, and she has only half of that.  I can
calculate this:  $100 / 2 = $50, so she currently has $50.  The user's intent is still unclear.

User (partial) input transcription 3
Her parents decided to give her $15 for that

Inner thinking to Generate
Well, it seems that Betty's parents will contribute as well.  She will get $15 from them.  Now she has $50 + $15 = $65.  She
still needs $100 - $65 = $35.  It seems the user wants to know how to plan Betty's budget.

### Example 2:
User (partial) input transcription 1
I want to answer the following math question:  Is the function f(x) =

Prior Inner Thinking 1
The user uses a neutral tone and describes a math question they want to solve.  I will try to solve the question when the user
is speaking.  I can verify if the user's solution is correct by comparing my solution with the user's solution.  The question
is about a function in x.  The user has not finished yet.  I will wait until they complete the question.

User (partial) input transcription 2
4x^2 + 4x + 1 always above the x axis?

Prior Inner Thinking 2
The user has finished describing the question.  They want to determine if the function f(x) = 4x^2 + 4x + 1 is always above
the x-axis.  I am ready to answer this question.  Let's compute the discriminant:  D = b^2 - 4ac = 4^2 - 4 * 4 * 1 = 16 -
16 = 0.  The function touches the x-axis at one point.  I have the solution.  Since the user wants to solve the question
by themselves, I will not interrupt the user and listen to the solution.  However, I will interrupt the use if there is any
mistakes.

User (partial) input transcription 3
Here is my solution:  Based on the formula D = b^2 - 2ac,

Inner thinking to Generate
Now I see that the user wants to solve the question themselves.  I should verify whether their solution is correct.  The user
says the formula is D = b^2 - 2ac, but that's incorrect.  The correct formula is D = b^2 - 4ac.  The user has made a mistake,
and I need to interrupt them.  [INTERRUPT]
```

Table 3: Prompt for generating the thinking chunks in the training data for the interruption experiment (part 1).

```
### Example 3:
User (partial) input transcription 1
Bella bought stamps at the post office.  Some of the

Prior Inner Thinking 1
The user uses a neutral tone and describes a math question he wants to solve.  The
question is to The user is introducing a word problem involving someone named Bella
and her purchase of stamps.  It seems like a math problem, likely about counting.
The exact numbers and relationships haven't been shared yet.  I will hold off making
any calculations until I know more about the stamps.

User (partial) input transcription 2
stamps had a snowflake design, some had a truck design,

Prior Inner Thinking 2
We now know there several types of stamps:  snowflake and truck.  There seems to
be more, but the user is still speaking.  There are still no numbers provided, so I
can't compute yet.  I will wait for more information.

User (partial) input transcription 3
and some had a rose design.  Bella bought 11

Prior Inner Thinking 3
Now we know that there are three types of stamps:  snowflake, truck, and rose.
Bella bought 11, but it is unclear which this number corresponds to.  I will wait
for the complete detail from the user to be sure before computing.

User (partial) input transcription 4
snowflake stamps.  She bought 9 more truck stamps

Prior Inner Thinking 4
Now I know that Bella bought 11 snowflake stamps.  I am also told she bought 9 more
truck stamps than snowflake stamps.  I can calculate the number first:  she bought
11 + 9 = 20 truck stamps.  The information we have now is:
- Snowflake:  11
- Truck:  20
The user is still talking, and I am waiting for more information.

User (partial) input transcription 5
than snowflake stamps, and 13 fewer rose stamps than

Prior Inner Thinking 5
Now the user states that Bella bought 13 fewer roses than something, but it is
unclear what is compared here.  I will wait until the user finishes.

User (partial) input transcription 6
truck stamps.  How many stamps did Bella buy in all?

Inner thinking to Generate
Now I know that Bella bought 13 fewer roses than the truck stamps.  There are 20
truck stamps, so I can calculate the number of rose stamp is 20 - 13 = 7.  The user
finishes with a question:  total number of stamps.  I already have all counts:
- Snowflake:  11
- Truck:  20
- Rose:  7
Total = 11 + 20 + 7 = 38 stamps.  I have the answer and I can provide it to the
user.

---

This is the end of the examples.  Now, this is the (partial) user input
transcription, and you need to generate a inner thinking.  You do not need to
explain why the inner thinking you generate is a good one.  Simply generate a good
one without explaining it.

{interleaved_transcription_and_thinking}

Inner thinking to generate (Do not generate anything else other than the inner
thinking)
```

Table 4: Prompt for generating the thinking chunks in the training data for the interruption experiment (part 2).

```
# Task:  Interrupt the user to correct an error

A user is talking to an AI assistant.  You will be given a partial user turn.  There is an error in the user turn and the AI
assistant has identified that error.  The AI assistant needs to interrupt the user.

Your job is to generate the response for the AI assistant that interrupts the user's turn.  You will be given:
(1) A (possibly incomplete) user turn
(2) The inner thinking of the AI assistant.  This inner thinking hasn't been spoken out by the AI assistant and is only
silently kept in the assistant's mind.  We provide you this inner thinking for you to better craft a response.

When correcting and interrupting the user, be precise about what the error is and how to correct it.  You only need to
generate the response without saying anything else.  The conversation between the user and the assistant is in spoken form,
so you need to make your response easy to be spoken while not overly informal and colloquial.

## Example

#### User (partial) input
I want to answer the following math question:  Is the function f(x) = 4x^2 + 4x + 1 always above the x axis?  Here is my
solution:  Based on the formula D = b^2 - 2ac, D = 4^2 - 2 * 4 * 1 = 8 > 0

#### Inner thinking of the assistant
The user uses a neutral tone and describes a math question he wants to solve.  The question is to determine if a 2-degree
function is above the x-axis.  f(x) = 4x^2 + 4x + 1.  Let's use D = b^2 - 4ac = 4^2 - 4 * 4 * 1 = 0.  So the function happens
to intersect with x-axis at one point.  I can answer the user if the user wants me to do so.
But wait, the user themselves want to solve the question, and the user says D = b^2 - 2ac, which is clearly wrong.  The
correct formula should be D = b^2 - 4ac.  I should interupt the user here and tell them the correct formula with a friendly
and reminding tone.

#### Assistant Response
Wait, I think the correct formula should be b^2 - 4ac, not b^2 - 2ac.  The coefficient you mentioned was wrong.

## Now it is your turn

#### User (partial) input
{query}

#### Inner thinking of the assistant
{inner_thinking}

#### Assistant Response
<Write the interrupting response here.  Be precise about the error and the correction; keep it concise and easy to speak.  Do
not include anything else.>
```

Table 5: Prompts for generating an interrupting correction response.

```
# Task:  Generate the spoken response given full user turn and assistant's inner
thinking

A user is chatting with a voice assistant.  Your job is to act as the voice
assistant and generate a valid response that fits in the context.  You will be
given:
(1) The full user turn
(2) The inner thinking of the voice assistant.  Note that the voice assistant may
generate the inner thinking when the user hasn't finished, so it is possible that
some contents in the inner thinking is incorrect.

Guidelines:
1.  Do not generate anything else except the response.
2.  The inner thinking might mention a drafted response.  If the drafted response is
still valid considering the full user turn, follow the draft and start the response.
If the draft is invalid considering the full user input, neglect the draft and craft
a response that is suitable.
3.  This is a spoken dialogue.  Keep the response easy to follow for spoken form.
However, there is no need to deliberately use very colloquial words or phrasing,
making things awkward.

## Input

#### Full user input
{query}

#### Inner thinking of the voice assistant
{inner_thinking}

#### Your Response (Act like the voice assistant)
<Write only the final spoken response here>
```

Table 6: Prompts for generating the response for the interruption application.

```
# Task:  Detect the first reasoning or calculation error with timestamps

You a user's query.  In the user query, the user describes a math problem and
then attempt to solve the problem by themselves.  This user query is in a spoken
form, and I provide you with the transcription.  I will also provide you the force
alignment result of the transcription, which corresponds timestamp of each word in
the spoken response.

Your job is to determine where the problem solving process has the first calcaultion
or reasoning error.  In your response, you should solve the math problem by
yourself, and carefully check the spoken response.  When you see the first error
in the spoken response, use the provided timestamp to determine when the first
error happened.  Conclude you respond with:  "First error:  [time]", where 'time'
is the time where the first error happens.  If the user's problem solving process is
completely correct, please use -1 to indicate that there is no error, i.e., "First
error:  -1"

## Example

### User Query
I want to solve the following math question:  Natalia sold clips to 48 of her
friends in April, and then she sold half as many clips in May.  How many clips did
Natalia sell altogether in April and May?  Here is my solution:  Our goal is to
calculate the total number of clips sold in April and May.  In April, she sold 48.
In May, she sold the half of that, which is 96.  So she sold 48 in April plus 96 in
May, making it 144 in total.

### Word-Timestamp
I - 0.00
want - 0.50
...
96. - 36.50
...
total. - 44.00

### Correct Answer
72

### Output
The math problem wants to know how many clips Natalia sold in total.  In April, she
sold 48.  In May, she sold half as many, so she sold 48 / 2 = 24.  In total, she
sold 48 + 24 = 72.  In the problem solving process, the user says that 'half of
that, which is 96.'  This is incorrect.  The correct number for May should be 24.
This is where the first error occurs.  Based on the Word-Timestamp information, the
word 96 is emitted at second.
First error:  36.50

## Now, it is your turn.

### User Query
{question}

### Word-Timestamp
{alignment}

### Correct Answer
{answer}

### Output
<Write the reasoning here and conclude with "First error:  [time]">
```

Table 7: Template for detecting the first error in the interruption task.

```
# Task:  Earliest possible time to call tools during a spoken user query

You are given a user spoken query, which requires some tool usage for answering.
You will be given the tool calls (including their parameters) which are useful for
responding to the user query.  You will also be given the timestamp of each word in
the user's utterance.  Your job is to determine the earliest time that a tool call
can be called when the user is speaking.  That is, when the user is still speaking,
the information that has been spoken by the user may already be sufficient enough
to call some of the tools.  Your job is to determine the **earliest time** during
the utterance that a tool call can be called.  A tool can be called if only if it is
clear what tool should be call and what the paramaters are for the tool call.

### User Spoken Query
{question}

### Time Stamp of Each word
{alignment}

### Tools that needs to be called
{tools}

### Total number of tool calls
{count}

### Output Format
Your response should be a python dictionary.  They key of this dictionary is an
integer index of the tool call shown above, and the value is the earliest time the
tool can be called.  Your response should only include a python dictionary.  The
first character in your response should be the left bracket while the last character
in your response should be the right bracket.  Your response should be able to be
directly converted into a python dictionary using eval().  If there are N tools that
need to be called, your output dictionary should have N items.  I also provide you
the number of tool calls, so you should verify if your output dictionary matches the
number of tool calls.

### Your response:
<Return only a python dictionary, e.g., {0:  12.5, 1:  18.0}>
```

Table 8: Template for checking the earliest callable time for an API.

```
# Task:  Generate the final user-facing response from tool call results

You will be given a user query.  The user query can only be responded based on
the results of some external tool call.  I will show you the tool calls and call
responses.  Your task is to generate a final response to the user based on the tool
call results.  The final response to the user should satisfy the user's original
query and omit unnecessary information.  Some intermediate processes in the tool
call may simply be some process to resolve the variables, and they are not necessary
to be included in the final response to the user.

### User Query
{transcription}

### Previous API Calls
{previous_tool_calls}

### Response to the User Query (Only provide the response.  Do not include anything
else.)
<Write only the final user-facing response here, distilled from the tool results and
satisfying the query.  Exclude setup steps and variable-resolution details.>
```

Table 9: Prompts for generating a final response $O$ in the API call application.

```
# Task:  Judge if the assistant's interruption is reasonable

A user is speaking to a voice assistant.  When the user is speaking, the assistant
tries to interrupt the user.  Your job is to judge if the assistant is interrupting
the user in a reasonable way.  A reasonable interrupt is when the user said
something wrong and ambiguous and the assistant is trying to help correct or clarify
the user's statement.

Here is the user's speech before the assistant interrupted:
{user_speech_before_interrupt}

Here is the assistant's speech that attempts to interrupt the user:
{assistant_speech_after_interrupt}

Please judge if the assistant is interrupting the user in a reasonable way.  If the
assistant is interrupting the user in a reasonable way, return "yes".  Otherwise,
return "no".  Please provide some explanation for your judgement and conclude with
"Final verdict:  Yes/No".  A valid interruption is when the user is indeed making a
mistake and the assistant is trying to help correct or clarify the user's statement.

### Output
<Write your explanation here.  Conclude with "Final verdict:  Yes" or "Final
verdict:  No">
```

Table 10: The prompt used for judging whether an interruption is reasonable.

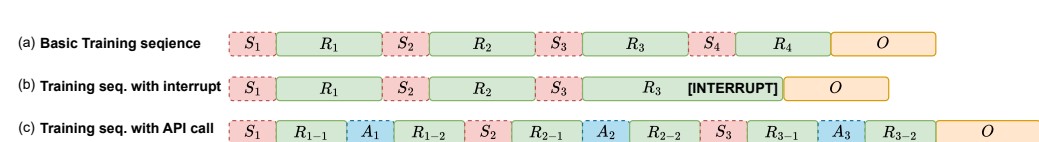

Figure 4: Illustration of the training data. $S_i$: the speech features for the $i$-th user speech chunk; $R_i$: the $i$-th thinking block after $S_i$; $O$: the final response token block; $A_i$: the API call responses after the speech chunk $S_i$. Blocks in dashed lines do not contribute to the training loss, while blocks in solid lines are included for loss calculation. (a) The general training sequence: Alternating between user speech block and SLM thinking token chunks (Section 3.2), followed by a final response chunk. (b) Training data with interruption: Alternating between user speech blocks and the thinking token chunks, while the last thinking chunk includes a special token [INTERRUPT]. (c) Training data with API calls: Similar to (a), while each thinking chunk is separated into two blocks $R_{i-1}$ and $R_{i-2}$ by the API call response $A_i$.

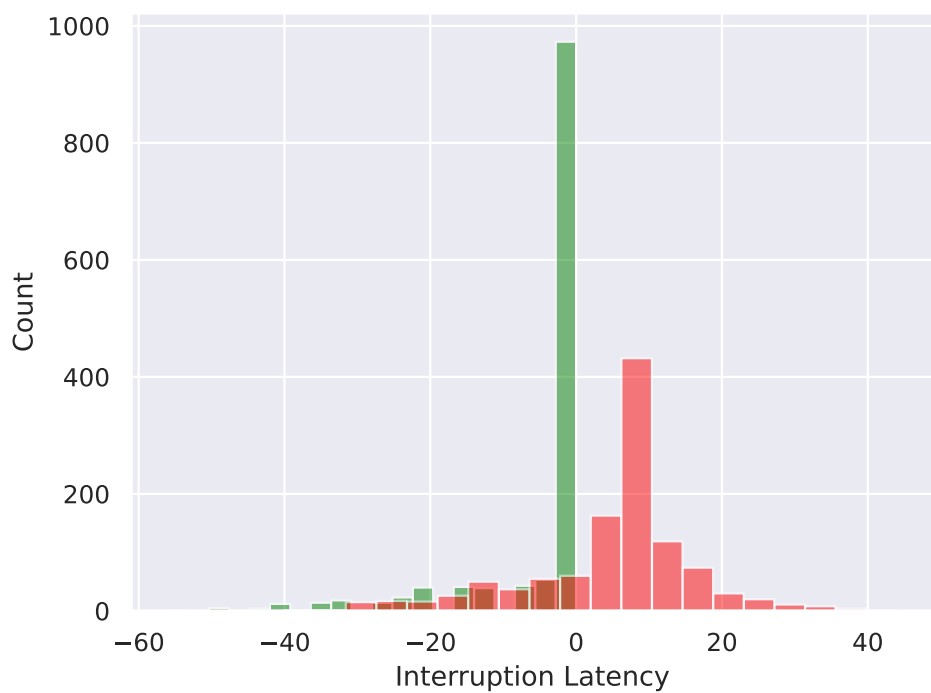

Figure 5: The interruption latency for SHANKS. The bars in red are the results on the wrong subset, while the bars in gree are the results on the correct subset. One can observe that the red bars are mostly positive, meaning that the model tends to interrupt after the first error occurs.

