# OpenReview forum: "Shanks: Simultaneous Hearing and Thinking for Spoken Language Models"
_ICLR.cc/2026/Conference — ICLR 2026 Conference Withdrawn Submission_

### Official Review · Reviewer_WjYh · 2025-10-25

**Soundness:** 2
**Presentation:** 3
**Contribution:** 2
**Rating:** 2
**Confidence:** 5

**Summary:**

This paper introduces SHANKS, an inference framework to enable SLMs to perform thinking while listening. The paper focuses on two tasks. 1) Interrupting the user when the model detects a mistake in the user's reasoning. 2) Making simultaneous API calls while the user is still speaking. Experimental results demonstrate that SHANKS can effectively handle both tasks.

**Strengths:**

- The paper is well-written and easy to follow.
- Experimental results demonstrate the effectiveness of SHANKS on the proposed two tasks.

**Weaknesses:**

- Over-claiming. The title “thinking while listening” is broad and ambitious, but the paper only addresses two specific tasks. A more accurate title might be “SLM for Educational Error Correction ...” It would also be more compelling to see an SLM with more general “thinking while listening” capabilities.
- Questionable setting. For the API-call scenario, it is unclear whether users would actually care about API-call latency. Users may prioritize task success rate over latency. Furthermore, some complex tasks may require several minutes to complete, while user queries may finish very quickly (e.g., in 10 seconds), making the latency reduction be trivial.
- Lack of Self-Containment. The paper fine-tunes a thinker-talker SLM but does not discuss how the talker is adapted to perform speech generation. In Qwen-2.5-Omni, the talker’s input includes hidden states from the thinker. The SHANKS approach introduces additional reasoning tokens, which may affect the hidden states of subsequent response tokens. This suggests that adapting the talker may not be trivial. Additionally, the evaluation appears to be performed only on the thinker’s text output, effectively reducing the SHANKS pipeline to a speech-to-text LLM.
- Lack of Real-Time Latency Evaluation from the User’s Perspective. The time for an API call is not a fixed number, so if the API call gets stuck, the user may never get the response.

**Questions:**

- In scenario 1, when the model interrupts the user, can the user interrupt back? Does the model continue listening to the user’s speech after the interruption? Can the model handle multi-turn dialogues?
- In Table 1, why is the valid interrupt ratio typically lower than the interrupt ratio in the “wrong” subset? Are there scenarios when an interruption occurs in the wrong subset but it is invalid?
- According to Section 5.5, in the “SHANKS + call-after-listen” setting, the early call mechanism is the same as in the “SHANKS” setting. Why, then, is the early call accuracy in “SHANKS + call-after-listen” higher than in “SHANKS” as shown in Table 2?

---

> ### Author Response · Authors · 2025-11-26
> **Response to Reviewer WjYh (1/2)**
>
> We would like to thank the reviewer for the detailed and human-written review. It is valuable to see human-written reviews these days. We respond to the weaknesses as follows.
>
> > Over-claiming. The title “thinking while listening” is broad and ambitious, but the paper only addresses two specific tasks. A more accurate title might be “SLM for Educational Error Correction ...” It would also be more compelling to see an SLM with more general “thinking while listening” capabilities.
>
> We appreciate the reviewer’s concern about the scope suggested by the title. Our intention in using “thinking while listening” was to highlight the general framework we introduce, which is not restricted to a single application, but can, in principle, support a variety of tasks: from dialogue-style interactions (Setting 2\) to interrupting and assisting during reasoning-intensive user speech (Setting 1).
>
> At the same time, we agree that the current empirical validation focuses on two concrete tasks. This is largely due to the novelty of the “thinking while listening” setting: to our knowledge, there are no existing benchmarks, datasets, or standard evaluations for this paradigm. As a result, we had to design and construct tasks and evaluations from scratch, which makes it challenging to cover a broader range of applications within a single paper. Our goal in this work is therefore to provide a proof of concept for the proposed framework and to demonstrate its potential, rather than to claim an exhaustive treatment of all possible “thinking while listening” capabilities.
>
> In light of the reviewer’s comment, we are open to adjusting the title to make the proof-of-concept nature more explicit. For example, we would be happy to change it to: “**Thinking While Listening for SLMs: A Proof of Concept on Two Tasks.**”
> We hope this addresses the concern while still accurately reflecting the generality of the proposed framework.
>
> > Questionable setting. For the API-call scenario, it is unclear whether users would actually care about API-call latency. Users may prioritize task success rate over latency. Furthermore, some complex tasks may require several minutes to complete, while user queries may finish very quickly (e.g., in 10 seconds), making the latency reduction be trivial.
>
> We agree that task success rate should be prioritized over latency, and **we propose an alternative, SHANKS+call after listening, which is shown in Table 2 in the paper. This method shows that high task success rate and low latency can be achieved at the same time.**
>
> > Lack of Self-Containment.
>
> In the paper, we said that fine-tuning the talker should be straightforward. This is because we just need to pass the thinker’s output embedding corresponding to the non-thinking tokens to the talker, and we only need to fine-tune the talker model to recognize the thinker’s output embedding after we fine-tune it for SHANKS. We outline this procedure in the 6th paragraph in Section 2\. However, this is not the focus of the paper, so we omit the details.
>
> In fact, our paper does not fine-tune the talker model. We explicitly say that we only focus on how to train the thinker model in the last paragraph in Section 2\. The reason, while not explicitly revealed in the paper, is that Qwen-2.5-Omni does not release the speech tokenizer, so it will be hard to prepare the training data to fine-tune the speech output of the talker model. This is also mentioned by Reviewer rFkn. In our revision, we will explicitly say that we do not fine-tune the talker model and only focus on the speech-in-text-out part, i.e., the thinker part.
>
> The reviewer might have a follow-up question about how we calculate the speech output latency in the experiments. We answer it in advance here. Here, we use the estimated latency from the technical report of Qwen-3-Omni (Table 2 in the paper), while adding additional latency due to the thinking generating the thinking tokens. Note that while we directly use the numbers from Qwen-3-Omni, the model of Qwen-3-Omni is actually larger than Qwen-2.5-Omni, so the latency we estimated in our paper may overestimate the real latency.

---

> > ### Author Response · Authors · 2025-11-26
> > **Response to Reviewer WjYh (2/2)**
> >
> > > Lack of Real-Time Latency Evaluation from the User’s Perspective. The time for an API call is not a fixed number, so if the API call gets stuck, the user may never get the response.
> >
> > We additionally evaluate the first-package latency as follows. The latency is calculated by considering the following three factors: (1) the time to generate the (remaining) thinking tokens after the user finishes the speech, and (2) the time to generate the first block of the text output from the thinker, which can be used to generate the speech using the talker, and (3) the time for the talker and speech decoder to generate the speech output. As mentioned in the previous response, (3) is estimated using the latency provided by Qwen-3-Omni. (1) and (2) are estimated by the token-generation speed on a single A100 GPU.
> >
> > |  | Latency (second) |
> > | :---: | :---: |
> > | SHANKS | 1.18 |
> > | Call-after-listen | 4.41 |
> > | SHANKS+call-after-listen | 1.96 |
> >
> > The above evaluation does not consider the situation that the API call may get stuck. This is because we want to focus more on the method we propose, so we simplify the situation to make the evaluation and comparison cleaner. If we consider the case when API calls get stuck, no matter which method we use, e.g., SHANKS or call-after-listening, the user may never get the response.
> >
> > > In scenario 1, when the model interrupts the user, can the user interrupt back? Does the model continue listening to the user’s speech after the interruption? Can the model handle multi-turn dialogues?
> >
> > No. In our current scenario, the interaction ends once the model interrupts the user. The user cannot interrupt back. This is because Qwen-2.5-Omni is not a full-duplex SLM, so it cannot listen to the user when it is interrupting the user. The current model does not consider multi-turn dialogues, but this is rather a limitation of the current evaluation rather than the method itself.
> >
> > \> In Table 1, why is the valid interrupt ratio typically lower than the interrupt ratio in the “wrong” subset? Are there scenarios when an interruption occurs in the wrong subset but it is invalid?
> >
> > The valid interrupt ratio is typically **higher** on the wrong subset. We do not fully understand the first part of the question and hope the reviewer can clarify your original intent. Thank you very much. About your second question, there are indeed cases when the interruption occurs in the wrong subset that is invalid. An interesting and quite common case is when the model interrupts the user when the user hasn’t finished the complete calculation. For example, the user may be saying, “*one hundred plus twenty equals one (hundred twenty)*”, but our chunking of $t\_{chunk}$ splits before the last two words. When this happens, our model may falsely interrupt the user by saying “*one hundred plus twenty should be 120, not 1\.*” This is one of the limitations of SHANKS, due to its chunking nature. We will include some discussions on the limitations of SHANKS in the revision, and we will add the above discussions in Section 4.5. Thank you for asking this question.
> >
> > > According to Section 5.5, in the “SHANKS \+ call-after-listen” setting, the early call mechanism is the same as in the “SHANKS” setting. Why, then, is the early call accuracy in “SHANKS \+ call-after-listen” higher than in “SHANKS” as shown in Table 2?
> >
> > This may be because, in the combined method, the model is trained on both the SHANKS format dataset and the call-after-listening dataset. As a result, the model is not exactly the same as SHANKS, even if they operate in the same way when the user is speaking. The higher accuracy may stem from the randomness during training or inference. After retraining the model three times using different random seeds, we found that the early call accuracy of SHANKS can range from 56.9 to 62.4, so the difference in the early call accuracy between SHANKS and SHANKS+call-after-listening may not be meaningful.

---

> > > ### Comment · Reviewer_WjYh · 2025-11-28
> > >
> > > I appreciate the authors' response.
> > >
> > > > In Table 1, why is the valid interrupt ratio typically lower than the interrupt ratio in the “wrong” subset? Are there scenarios when an interruption occurs in the wrong subset but it is invalid?
> > >
> > > For this part, the meaning is that, in the "wrong" subset part of Table 1, the scores of the "valid interrupt ratio" are typically lower than the "interrupt ratio". However, I believe the authors have already addressed this question.
> > >
> > > Although some limitations of the paper remain, I am willing to adjust my score slightly to acknowledge the authors' rebuttal.
> > >
> > > Note: There seems to be no "edit" button for the official review at the moment, so I cannot directly update my score. However, my intention is to change my Rating from 2 to 4.

---

### Official Review · Reviewer_rFkn · 2025-10-27

**Soundness:** 3
**Presentation:** 3
**Contribution:** 3
**Rating:** 6
**Confidence:** 5

**Summary:**

The paper proposes a pipeline for Speech-to-Text (presumably) multimodal LLMs that performs intermediate reasoning while listening, in contrast to most existing LLMs or SLMs that execute reasoning or thought processes only **after** receiving the full input. The proposed method demonstrates its effectiveness particularly in mathematical speech correction and tool-calling tasks, thereby validating the utility of the introduced pipeline.

**Strengths:**

Personally, I found this paper quite engaging. It is written in an accessible way, and the figures are intuitive and easy to understand. I was particularly impressed by how the proposed approach partially aligns with human-like reasoning patterns.

**Weaknesses:**

Rather than noting weaknesses, I have summarized a few personal questions and minor concerns below:

**Questions:**

1. **Use of LLM-as-a-Judge in Mathematical Tasks**
   The paper employs an LLM-as-a-judge framework to evaluate reasoning and interruptions in math-related tasks. As you know, reasoning tasks are inherently more challenging than general ones, and I still have slight concerns about whether current LLMs have reached a level of convergence sufficient for reliable evaluation. Perhaps I missed it, but I would like to know which API or model was used as the judge and whether its evaluations correlate with human judgments. In other words, I would like to see evidence that the LLM is capable of evaluating such novel tasks accurately.

2. **Clarification on the Speech Modality (Speech-to-Speech vs. Speech-to-Text)**
   From the figure, the architecture appears to have a speech-to-speech structure with a decoder attached. However, to my knowledge, *Qwen2.5-Omni* does not currently support speech-to-speech SFT, as the encoder of the codec module used for decoding is not publicly released. In my experience, even when tuning the core thinker module with LoRA (especially with full tuning), compatibility with the talker module often breaks, leading to generation failures. Should I understand the proposed system as *speech-to-text* instead? If so, it might be helpful to clarify or revise the figure and corresponding text. On the other hand, if it truly is *speech-to-speech*, I am curious how you addressed potential representation mismatches between the two modules that could degrade generation quality.

3. **Generation Stability in Variable-Length Interleaving**
   As I understand it, prior interleaving approaches, those that handle listening and reasoning concurrently or operate in a full-duplex manner (e.g., OmniFlatten, SyncLLM), typically generate a *fixed-length* assistant response chunk corresponding to each user input chunk to maintain generation stability. This fixed-chunk design, in my view, helps mitigate endless generation issues and allows manual truncation when necessary. In contrast, your method appears to train on *variable-length* reasoning trajectories aligned with the preceding input chunks. I am curious whether you encountered any generation instability (e.g., occasional repetition or degradation in 1 out of 100 samples, etc.) during this variable-length reasoning chunks generation.

4. **Comparing “Thinking While Listening” vs. “Thinking After Listening”**
   This is not a criticism but rather a personal curiosity. I personally believe that enabling reasoning or assistant speech *during listening*, as your pipeline does, represents the future direction of speech LLMs. From that perspective, its conceptual counterpart would be the conventional *turn-based* approach, where the assistant reasons and responds *after* the user’s utterance ends. While you did compare the two in the tool-calling experiment, I am particularly interested in how they differ in more reasoning-intensive tasks such as mathematics. Specifically, it would be fascinating to compare “thinking while listening” and “thinking after listening” pipelines in terms of reasoning accuracy, interruption prediction quality (i.e., whether interruptions are appropriate), and the validity of the interruption content. I understand that latency might be impossible to measure, and I do not view lower performance of the after-listening variant as a weakness, but including such an analysis could enhance the completeness and informativeness of the paper.

Overall, I found this paper very enjoyable and thought-provoking. Thank you for your work. If additional experiments addressing the above questions are provided, I would be happy to raise my score.

---

> ### Author Response · Authors · 2025-11-26
> **Response to Reviewer rFkn (1/2)**
>
> > Use of LLM-as-a-Judge in Mathematical Tasks
>
> Thank you for your question. We use GPT-4o as the LLM judge to determine if the interruption is valid. We did not include this in the paper, and we will update the revision to include this information. Given the high accuracy of GPT-4o on GSM8K (over 90% on the testing set of GSM8K), we think using GPT-4o as the judge in this task is reliable.
>
> > Clarification on the Speech Modality (Speech-to-Speech vs. Speech-to-Text)
>
> It is correct that we only perform speech-to-text fine-tuning on Qwen-2.5-Omni, and the reason is exactly like what you said: the speech encoder is not publicly available. And we did find that after fine-tuning the thinker, the talker cannot generate the speech properly. In Section 2 of our paper, we only state that fine-tuning the talker will be necessary, without discussing how to do so. Still, we believe that as long as we have the necessary resources to fine-tune the talker, adapting the talker should be quite straightforward. We will update our revision to clearly reveal this.
>
> In our Figure, we draw it as a speech-to-speech framework since we think that thinking while listening is more natural in speech-to-speech interaction. However, due to practical limitations of Qwen-2.5-Omni, we cannot demonstrate the complete speech-to-speech ability of the method. We will update the caption in Figure 1 to explicitly highlight that while SHANKS can be used in a speech-to-speech way, our paper only trains the speech-to-text part.
>
> Last, we would like to add some comments about the possibility of fine-tuning the talker model even without the speech encoder. Given a pair of input and output speech to train the complete speech-to-speech capability of Qwen-2.5-Omni, we can obtain the target speech tokens using the original Qwen-2.5-Omni by the following steps:
>
> 1. Transcribe the output speech into text.
> 2. Feed the speech input and text output into the thinker model, i.e., using teacher forcing for the output texts of the thinker model. This will give us the thinker’s output embedding when saying the content we want.
> 3. Feed the thinker output embedding to the talker to get the generated audio tokens for the target transcription.
>
> Eventually, we did not adopt this method to get the speech tokens for fine-tuning, as the above method requires forwarding the whole training dataset through Qwen-2.5-Omni, which requires significant resources. But the above pipeline should be workable.
>
> > Generation Stability in Variable-Length Interleaving
>
> Thank you for your interesting question. It is correct that SHANKS uses variable-length assistance turns as the reasoning process. To be more precise, we limit the assistant’s thinking chunk length to 320 tokens (the number of tokens that can be generated in $t\_{chunk}=4$ seconds on an A100). If the length of reasoning tokens exceeds this number, we truncate it. If it is less than 320, we also do not pad it to 320 tokens. Interestingly, we do observe that the model’s output degenerates in about 1% of the outputs; the degeneration is mostly repetition, and this cannot be solved by setting the repetition penalty during inference. We have not found a reason for this issue and have been unable to resolve it.
>
> We initially thought this was an issue due to the speech input of Qwen-2.5-Omni. However, we also found a similar issue in our cascade SHANKS, which indicates that even text input can have the same problem. Recall that in the cascade SHANKS, we utilize an ASR and an LLM pipeline; therefore, fine-tuning the LLM for SHANKS essentially involves training the LLM for multi-turn dialogue. Fine-tuning LLMs for multi-turn dialogue is a natural fine-tuning setting for LLMs, and variable-length assistant responses should be easy to train, but the LLM still degenerates.
>
> Finally, we think this may be more related to the model itself, as fine-tuning the same dataset on Llama-3.1, i.e., using Llama-3.1 as the backbone LLM for cascade SHANKS, we do not observe severe degradation. Why LLMs degenerate seems to remain an unresolved and underexplored research problem, and we are looking forward to seeing more research on this. We are willing to learn more feedback from the reviewer on this topic.

---

> > ### Author Response · Authors · 2025-11-26
> > **Response to Reviewer rFkn (2/2)**
> >
> > > Comparing “Thinking While Listening” vs. “Thinking After Listening”
> >
> > Thank you again for your question. We did not include the thinking after listening in the interruption experiment because if the model starts to think and respond after the listening ends, the user’s speech has already finished, and there is nothing to interrupt. So an alternative way to frame your question is more like, “*Can the model really know what the error is when they are given a complete user speech?*” We report some numbers here for Qwen-2.5-Omni:
> >
> > 1. Qwen-2.5-Omni’s accuracy on GSM8K is 91.6%. This is the accuracy when you give the model the question and ask it to answer it.
> > 2. When we give Qwen-2.5-Omni a question from GSM8K and a (possibly wrong) step-by-step solution, and ask the model if the solution is correct.
> >    1. For the correct subset, the model says the solution is correct in 98.2% of the time.
> >    2. For the incorrect subset, the model says the solution is correct in 37.4% of the time.
> >
> > So this means that for Qwen-2.5-Omni, answering the question is easy, agreeing on the correct answer is easy, but identifying errors from a wrong solution, even if we provide the complete (not truncated) incorrect step-by-step solution, is challenging. We also have an experiment to prompt Qwen-2.5-Omni in a multi-turn setting, where we first present the model with a question, ask it to answer, and in the third turn, we provide a wrong answer and ask whether it is correct or not. Still, the model performs similarly to situation 2.(b) as the above.
> >
> > Finally, we would like to thank you again for your review and for the warm gesture that indicated the possibility of raising the score. However, I personally do not feel like I provide sufficient responses that are worth raising the score. But I deeply enjoy the process of reading your positive comments and thoughtful questions, which reminds me of why I set up this project and what I can do better. The authors are willing to engage in further discussions and answer any follow-up questions, and we are sorry for the late reply.

---

> > > ### Comment · Reviewer_rFkn · 2025-11-27
> > >
> > > First of all, thank you very much for your kind and detailed response. Although it is somewhat disappointing that there is no clear correlation with human evaluation, I understand that evaluating such reasoning behaviors in this setting is inherently difficult using human eval, especially for this type of task.
> > >
> > > I also appreciate the clarification regarding modality (s->t vs s->s), and I personally found it very interesting that you mentioned the repetition issue is likely caused by the backbone model itself. As someone who has experimented with a somewhat similar form of variable interleaving (though not identical), this was an aspect I had overlooked, and your explanation was quite insightful.
> > >
> > > Finally, regarding the additional experiments you mentioned: I may not have fully captured it, so could you please clarify exactly which values in your paper correspond to the two metrics you measured in this additional analysis? In other words, I would appreciate it if you could indicate how these measured values map to the results reported in the existing “thinking while listening” experiments in Table 1.

---

> > > > ### Author Response · Authors · 2025-11-27
> > > > **Re: Reviewer rFkn**
> > > >
> > > > Thank you for your quick follow-up.
> > > >
> > > > About the additional experiment we mentioned in the previous response, it is in response to your previous comment on comparing thinking while listening and thinking after listening in the math reasoning (interruption) experiment. Our main point is that **the interruption experiments and all the metrics we show in Table 1 only make sense when the model interrupts when it is still listening**, so it is not possible to report those metrics under the "thinking after listening" scenario, as there is nothing to *interrupt* after the listening has ended.
> > > >
> > > > So, rather than reporting the interruption metrics, in the additional experiment results, we directly evaluate whether the model can reason correctly after the model has listened to the user's full solution.
> > > >
> > > > Please let us know if our clarification answers your question. If not, we are happy to discuss more on this. Thanks.

---

> ### Comment · Reviewer_rFkn · 2025-11-27
>
> Thank you for your response. I understand the situation, and it stemmed from my own misunderstanding. In any case, I believe the methodology can be applied in many useful ways, and I found it very inspiring. The reviewer’s response also addressed my questions to some extent. Therefore, I remain optimistic.
>
> P.S. I intend to increase the score to 8; however, for some reason, the edit button for the original review is not visible at the moment. I will update it as soon as it becomes available.

---

> > ### Author Response · Authors · 2025-11-28
> > **Re: Reviewer rFkn**
> >
> > Thanks for your response and for indicating that you will increase the score. Your positive comments and insightful feedback mean a lot to us.

---

### Official Review · Reviewer_nwfy · 2025-10-30

**Soundness:** 3
**Presentation:** 3
**Contribution:** 2
**Rating:** 2
**Confidence:** 3

**Summary:**

This paper proposes Shanks, a general inference framework that enables spoken language models (SLMs) to determine whether to interrupt the user's input speech when the user is making mistakes. Shanks starts the reasoning as soon as the input speech from the user is received. This way it reduces the time the user needs to wait after their speech is done, and also the user can correct their misunderstandings in their intermediate steps. Experimental results on math problems and task-oriented dialogues show substantial improvement over the baselines.

**Strengths:**

- The paper is clearly written and easy to follow.
- The core concept, “thinking before speaking,” is interesting. The extension of this idea to conversational AI, particularly through the proposed “interrupt” functionality, is creative. Such mechanisms could be beneficial for users studying logical or structured domains like mathematics or science, where step-by-step reasoning feedback is valuable.
- The authors also constructed a speech dataset derived from existing text-based reasoning datasets to train and evaluate their models. It would be helpful for the research community, but I'm not entirely sure how these datasets support future research beyond their current experiments.

**Weaknesses:**

- The core idea of this paper: "Shanks: Simultaneous Hearing and Thinking for Spoken Language Models" is already described in the other paper: "STITCH: Simultaneous Thinking and Talking with Chunked Reasoning for Spoken Language Models". In STITCH, the authors also had the same motivation as this paper, which is "...humans can think while speaking and propose STITCH-R".
The main difference might be the interruption function of SHANKS. The paper does not include direct experimental comparisons with STITCH models (STITCH-R and STITCH-S) despite the conceptual overlap. It is unclear what additional benefit or novelty SHANKS provides, considering that SHANKS performance on a task-oriented dataset performs worse than call-after-listen.

- I’m not convinced about the usefulness of SHANKS in other contexts, such as mental consulting or conversational counseling. In these settings, users often describe the main issue only at the end, and much of the earlier speech may be digressive or irrelevant. SHANKS does not explain how it identifies and prioritizes the most important context segments—it simply begins reasoning as soon as the input speech starts. This approach could cause significant computational overhead and wasted inference cycles without improving understanding.

- In educational contexts, premature or incorrect interruptions by SHANKS could lead to confusion or misunderstandings rather than improving interactivity. Some human experiments, like how humans actually find this useful, would help to strengthen the paper's results.

- While SHANKS shows strong performance improvements on math reasoning datasets, its results on task-oriented dialogues (e.g., flight or hotel booking scenarios) are mixed. In particular, the call-after-listen baseline often performs better than SHANKS itself. Moreover, user inputs in such scenarios can be long and continuously updated. For example, time-sensitive information like flight details may change quickly, so initiating the reasoning process too early can lead to outdated or irrelevant inferences.

**Questions:**

My questions and suggestions are listed throughout the strength and weakness sections.

---

> ### Comment · Reviewer_WjYh · 2025-11-12
>
> Couldn't agree more with weakness 1! STITCH and SHANKS have significant overlaps. If the papers come from the same group of authors, I don’t see a clear reason why they should be separated into two papers.

---

> ### Author Response · Authors · 2025-11-12
> **Clarification on the difference between STITCH and SHANKS (1/2)**
>
> We thank the reviewers for their review. Before we provide point-by-point responses later in the discussion period, we would like to clarify a central misunderstanding that appears in the comments from reviewers nwfy and WjYh.
>
> Reviewer nwfy states that "*The core idea of this paper: "Shanks: Simultaneous Hearing and Thinking for Spoken Language Models" is already described in the other paper: "STITCH: Simultaneous Thinking and Talking with Chunked Reasoning for Spoken Language Models". In STITCH, the authors also had the same motivation as this paper, which is "...humans can think while speaking and propose STITCH-R".*"
>
> **The above statement is wrong**.  SHANKS (this paper) is thinking while **listening** (to the user speech), while STITCH is thinking while **speaking** (the SLM's response). The above concept of SHANKS (thinking while **listening**) **did not appear in any part of STITCH**. This “*when the model thinks*” dimension is the core axis of difference, and the **“thinking while listening” regime explored in SHANKS does not appear in STITCH**.
>
> Because of this, the difference is not merely about adding an interruption function on top of an existing idea. Rather, SHANKS targets a different phase of the spoken interaction (the listening phase) and thereby enables different behaviors (e.g., responding or interrupting earlier because reasoning is already underway during user speech).
>
> Given this distinction, we respectfully disagree with the comment by Reviewer WjYh that “*STITCH and SHANKS have significant overlaps.*” If the reviewers believe there are specific sections, including the experiment settings or the method, in SHANKS that duplicate STITCH’s 'thinking while speaking' setting, we would very much appreciate **pointers to those concrete parts** so we can address them directly in future discussions or clarify them in the final version.

---

> > ### Author Response · Authors · 2025-11-12
> > **Clarification on the difference between STITCH and SHANKS (2/2)**
> >
> > We acknowledge that both the STITCH paper and our submission employ an unspoken (silent) CoT. However, silent CoT, i.e., letting the model reason without emitting tokens to the user, has already been widely explored in the broader literature on reasoning LLMs (e.g., Deepseek-r1 or OpenAI o1), so using it alone should not be taken to imply substantial overlap between STITCH and SHANKS. What matters here is where in the spoken interaction the silent CoT is placed and what behavior it enables. In STITCH, the silent CoT is executed during the model’s own speaking phase, after the user has finished, to make use of that output time. In our work (SHANKS), the silent CoT is executed while the user is still speaking, so that the model can decide earlier (e.g., to interrupt or to issue an API/tool call) before the user's turn ends. This difference in placement and the resulting early-action capability is the main distinction we want to clarify.

---

> ### Author Response · Authors · 2025-11-26
> **Resposne to Reviewer 2/2**
>
> > While SHANKS shows strong performance improvements on math reasoning datasets, its results on task-oriented dialogues (e.g., flight or hotel booking scenarios) are mixed. In particular, the call-after-listen baseline often performs better than SHANKS itself. Moreover, user inputs in such scenarios can be long and continuously updated. For example, time-sensitive information like flight details may change quickly, so initiating the reasoning process too early can lead to outdated or irrelevant inferences.
>
> We break down the questions into two parts:
>
> 1. **SHANKS is worth than call-after-listening**
>
>    It is true that SHANKS outperforms call-after-listening. However, the two methods can be combined to achieve a better performance, as shown in Table 2\. The combined method enjoys the early calling of SHANKS, which can lead to reduced latency, while also having the high accuracy of call-after-listening. The low accuracy of SHANKS, when used alone, is because we restrict SHANKS to make the API call within the $t\_{chunk}$, and once the $t\_{chunk}$ is over, we will stop the API call. This makes SHANKS unable to retry failed API calls when the user is still speaking. However, when SHANKS is combined with call-after-listening, the model will retry the failed APIs after the user has finished, thus improving the task accuracy.
>
>
>
> 2. **The information in the user speech may change quickly, and the earlier tool calls may be outdated or irrelevant**
>
>    This is a trade-off between latency and redundant inference cost. If we always call after the user finishes, there will always be latency. However, if we make API calls when the user is speaking, this will be useful when the user is consistent throughout the user's turn, while wasting the inference compute when the user changes their plan during their turn. Given this latency / inference-cost trade-off, the model deployer can determine by themselves whether the latency is more important while the inference cost is somewhat acceptable. Given the popularity of scaling the compute budget at inference time, we believe the inference cost for a possibly redundant thinking process is acceptable, and this will likely become more acceptable in the future as more advanced methods are used to reduce the cost of inference time computation.

---

> ### Author Response · Authors · 2025-11-26
> **Response to Reviewer (1/2)**
>
> We are grateful for the reviewer’s detailed feedback. We respond to the weakness and questions as follows.
>
> > The core idea of this paper: "Shanks: Simultaneous Hearing and Thinking for Spoken Language Models" is already described in the other paper: "STITCH: Simultaneous Thinking and Talking with Chunked Reasoning for Spoken Language Models". In STITCH, the authors also had the same motivation as this paper, which is "...humans can think while speaking and propose STITCH-R". The main difference might be the interruption function of SHANKS. The paper does not include direct experimental comparisons with STITCH models (STITCH-R and STITCH-S) despite the conceptual overlap. It is unclear what additional benefit or novelty SHANKS provides, considering that SHANKS performance on a task-oriented dataset performs worse than call-after-listen.
>
> Please refer to our earlier response on this weakness. As an additional note, in the second paragraph of Section 2, we have highlighted our difference with STITCH: SHANKS can think before the user has finished their turn, while STITCH can only think after the user has ended. Due to the above difference, they are used for different scenarios and are thus not directly comparable.
>
> > I’m not convinced about the usefulness of SHANKS in other contexts, such as mental consulting or conversational counseling. In these settings, users often describe the main issue only at the end, and much of the earlier speech may be digressive or irrelevant. SHANKS does not explain how it identifies and prioritizes the most important context segments—it simply begins reasoning as soon as the input speech starts. This approach could cause significant computational overhead and wasted inference cycles without improving understanding.
>
> It is correct that at the beginning of the user input, the information might be irrelevant, and reasoning based on these contents may be a waste of computing resources. When constructing the training data, if the current input chunk does not provide anything to reason, the target thinking chunk will be something like “there is not much information, and I will wait for more information”, as shown in example 3 in Table 4 in the Appendix. Ideally, the model will learn that if there is not much to think about, it will not think too much.
>
> We also agree with the reviewer that in other contexts, SHANKS’s thinking-while-listening might not always be useful. And this is why we deliberately select the interruption and tool-call during listening: These are the tasks where thinking while listening can really help. In our paper, we do not claim that SHANKS will be universally useful for all kinds of tasks. Our goal is to propose a way to achieve thinking while listening in SLM, which can be useful in some scenarios. Model deployers, who will be aware of what scenarios the model will be deployed to, can determine if their task of interest can benefit from SHANKS.
>
>
> > In educational contexts, premature or incorrect interruptions by SHANKS could lead to confusion or misunderstandings rather than improving interactivity. Some human experiments, like how humans actually find this useful, would help to strengthen the paper's results.
>
> We are grateful for the reviewer’s feedback on this. After some research, we do find mixed human studies on whether it is better to interrupt in the middle and or correct after the students have finished speaking. Considering that, the educational setting does not seem to be a good example of where SHANKS can be applied. Still, the concept of making an interruption once an error is made is a natural human behavior, and our goal is to use SHANKS, i.e., thinking while listening, to enable the SLMs to interrupt, just like humans. In our revision, we will modify the relevant contents by removing the educational setting and changing it into the other-initiated correction (correction made not made by the speaker, but by the listener) in natural dialogue

---

> > ### Author Response · Authors · 2025-11-27
> > **Asking for Response from Reviewer WjYh and nwfy**
> >
> > Dear Reviewer WjYh and nwfy,
> >
> > Thank you for your review. Previously, you indicated in the review or comment that our paper has a significant overlap with another paper, STITCH. However, we do not see concrete and verifiable evidence to support the above claim in the review or comment. **We thus asked for your clarification two weeks ago, but we have not received your responses**. We would like to ask you to provide justifications for these points if you believe the above statement is correct. If you cannot provide evidence to support this, please acknowledge this and/or modify your original review to reflect this.
> >
> > We want to emphasize that regardless of the final acceptance decision, the public comments posted on OpenReview will continue to affect how our work is perceived and evaluated by the community. We welcome constructive discussion, including criticism, when it is grounded in concrete evidence and well-supported reasoning. However, publicly visible statements that are incorrect or based on misunderstandings can cause lasting harm to a paper’s reputation, which we do not believe aligns with the intended purpose of OpenReview at ICLR. Ensuring the accuracy and appropriateness of public comments is therefore important not only for this submission but also for maintaining the fairness of the open reviewing process.

---

> > > ### Comment · Reviewer_WjYh · 2025-11-28
> > > **Clarification**
> > >
> > > I appreciate the authors' response. Sorry for the late reply. First, I would like to clarify that **I did not take this into account in my original review**. This is why I did not make this claim in my original review. However, **I friendly present the following since the authors would like further clarification.**
> > >
> > > Central response: The authors treat "thinking while listening" and "thinking while speaking" as if they are fundamentally different research problems. I respectfully disagree since both are instances of the same general problem:
> > >
> > > "How can we enable SLMs to perform unspoken reasoning in parallel with speech processing?"
> > >
> > > The answer in both papers is identical: chunk the speech and interleave reasoning. The only difference is whether we should conduct unpoken reasoning in the listening phase or the speaking phase. Therefore, it is not that SHANKS duplicates STITCH's exact experimental setting. It is that STITCH and SHANKS share the identical core methodology and merely apply it to different phases of the interaction.
> > >
> > > > "We acknowledge that both the STITCH paper and our submission employ an unspoken (silent) CoT. However, silent CoT... has already been widely explored in the broader literature... so using it alone should not be taken to imply substantial overlap."
> > >
> > > My understanding of the overlap is not just about using silent CoT. It is about both papers include:
> > > 1. Chunking speech into fixed-duration segments
> > > 2. Generating reasoning after each chunk
> > > 3. Interleaving reasoning with speech chunks
> > > 4. Using the same training data construction pipeline: they use GPT-4o to generate reasoning, and use GPT-4o-mini-TTS to synthesize the speech.
> > > 5. Evaluating on similar datasets (GSM8K, math reasoning)
> > > 6. Additionally, Figure 1 in both papers is strikingly similar in style.
> > >
> > > > "In our work (SHANKS), the silent CoT is executed while the user is still speaking, so that the model can decide earlier (e.g., to interrupt or to issue an API/tool call) before the user's turn ends. This difference in placement and the resulting early-action capability is the main distinction."
> > >
> > > I see the early-action capability is an application-level feature, not a methodological innovation, as it simply involves changing the unpoken CoT's position from SLM's speaking phase to the listening phase.
> > >
> > > I hope the authors can understand my clarifications. Thanks!

---

> > > ### Comment · Reviewer_nwfy · 2025-11-28
> > >
> > > Sorry for the late reply. I appreciate the response. I fully agree with the reviewer WjYh, and I'll also make a further clarification. The reason I view this work as substantially overlapping with STITCH is that both papers introduce the same core mechanism: chunked, unspoken chain-of-thought reasoning inserted into the spoken-language modeling pipeline and differ mainly in when this hidden reasoning is generated. Both papers rely on the same reasoning datasets (evaluation on GSM8K and training using Tulu-3 reasoning corpus), the same GPT4o generated CoT traces, and the same architectural pattern of alternating reasoning/text/speech tokens. As a result, the contribution of SHANKS looks like a reordering of an otherwise identical method rather than a substantively new contribution.
> > >
> > > Authors also admitted "SHANKS alone performs worse than a simple call-after-listening baseline" on the task-oriented dialogue scenario and needs to be combined with the call-after-listening baseline. As authors also mention "we do not claim that SHANKS will be universally useful for all kinds of tasks. Our goal is to propose a way to achieve thinking while listening in SLM, which can be useful in some scenarios", which confirms the method's applicability is limited.
> > > I would appreciate it if the authors could demonstrate SHANKS's value in a broader and more diverse set of use cases, and provide clearer evidence that the proposed approach yields robust, practical benefits across realistic spoken-interaction settings.
> > >
> > > Hence, I keep my score.

---

> > > > ### Author Response · Authors · 2025-11-28
> > > > **Response to Reviewer nwfy**
> > > >
> > > > Thank you for your follow-up. We ask you to kindly refer to the comments we replied to reviewer WjYh for our responses to the claimed similarity, and we want to add another correction to an incorrect statement in the previous comment:
> > > >
> > > > > Both papers rely on the same reasoning datasets (evaluation on GSM8K and training using Tulu-3 reasoning corpus)
> > > >
> > > > This is incorrect. SHANKS is not trained on Tulu-3 dataset.
> > > >
> > > > Additionally, the following statement:
> > > > > same architectural pattern of alternating reasoning/text/speech tokens
> > > >
> > > > also seems odd since SHANKS only interleave the input speech features and reasoning tokens.
> > > >
> > > > >  As authors also mention, "we do not claim that SHANKS will be universally useful for all kinds of tasks. Our goal is to propose a way to achieve thinking while listening in SLM, which can be useful in some scenarios", which confirms the method's applicability is limited.
> > > >
> > > > Saying a method's applicability is limited is somewhat different from our original sentence, which says that we do not claim that our method is universally useful. Our above statement is to highlight that we humbly admit the possible limitations of our methods. **We, as responsible researchers, are obliged to honestly point out possible limitations of our method in order to avoid hypes and overclaim. We also want to remind ourselves and our readers that no method is universal or perfect; all methods have limitations.** We appreciate the reviewer's expectation of seeing more interesting applications beyond what we have done in the paper. However, due to the limited content of a 10-page paper, many more applications are beyond the current paper's capacity.
> > > >
> > > > Last, we deeply appreciate your comments and responses. We have a lot of inspiration from your feedback.

---

> ### Author Response · Authors · 2025-11-28
> **Response to reviewer's clariifcation**
>
> Thank you very much for the detailed response and for mentioning that this was not taken into account in your original review. We appreciate your point-by-point breakdown, which helps us discuss the topic more easily. We respond to your explanation on the significant overlap of the two papers as follows. First, we would want to emphasize that we understand that the judgment of **significant** overlap is highly subjective, so we fully respect the reviewer for saying this and do not mean to argue with the reviewer on this, but rather provide our perspectives on why we do not think the above points justify saying the two papers have significant overlap. We list our response as a friendly discussion in response to the reviewer's friendly response.
>
> > The answer in both papers is identical: chunk the speech and interleave reasoning. The only difference is whether we should conduct unpoken reasoning in the listening phase or the speaking phase. Therefore, it is not that SHANKS duplicates STITCH's exact experimental setting. It is that STITCH and SHANKS share the identical core methodology and merely apply it to different phases of the interaction.
>
> The summary is correct. As the reviewer says, the two papers differ in when the model reasons.  This difference already marks a significant difference from all prior works. All past methods, no matter the reasoning, LLMs or SLMs, including STITCH, can only reason when the input is finished, while SHANKS enable a new interaction way between humans and foundation models. This is an important contribution of this paper that is not seen in any previous paper. **Saying the reasoning time is changed from speaking to listening somewhat underplays the implication and contribution of thinking during the input time of SHANKS.**
>
> > My understanding of the overlap is not just about using silent CoT. It is about both papers include:
>
>
> > 1. Chunking speech into fixed-duration segments
>
> Chunking speech into fixed-duration segments is a very common approach in speech processing. SHANKS chunks the input speech, which is quite common in real-time speech-input systems like streaming S2S translation or streaming ASR.  STITCH chunks the output speech, where chunking and interleaving speech and text are quite common in recent interleaved SLMs.
>
> > 2. Generating reasoning after each chunk
> > 3. Interleaving reasoning with speech chunks
>
>
> The two points are highly similar and related, so we address them together. Interleaving two concurrent streams of signal is a common approach in signal processing, which is called *time multiplexing*. This is not a characteristic unique to the two papers, but rather one that has already been explored in many prior works. SHANKS interleaves the **input speech** with model-generated reasoning, while STITCH interleaves the **output speech** with model-generated reasoning. For SHANKS, since we need to reason about the input, the model should generate the reasoning after the input speech chunk. This is a necessary, intuitive, and reasonable design. Similarly, it appears that reasoning after the speech chunk is a necessary design for STITCH, since the model needs to generate reasoning before generating the output; this is a standard in reasoning LLMs.
>
>
>
>
> > 4. Using the same training data construction pipeline: they use GPT-4o to generate reasoning, and use GPT-4o-mini-TTS to synthesize the speech.
>
> **The two papers use very different data construction pipelines**, and the only thing they share is the models they use to generate the data. Please refer to the data construction pipeline of the two papers, including the prompts the two papers used, the way to generate the chunked reasoning, and the filtering method. The two papers both use the two OpenAI models to generate the reasoning and speech. Countless papers use GPT-4o for generating training data due to its accessibility, stability, and high quality. Similarly, GPT-4o-mini-TTS is a very strong TTS model. **Simply because two papers use the same model to generate the training data does not make them have the same data construction pipeline. This is a significant overclaim.**
>
> > 5. Evaluating on similar datasets (GSM8K, math reasoning)
>
> This seems like an oversimplification. SHANKS evaluates on GSM8K for interruption and on ComplexFuncBench for task-oriented dialogue with tool call, while STITCH evaluates on AddSub, SVAMP, MultiArith, GSM8K, SinglEq, LlamaQA, WebQA, TriviaQA, and AlpacaEval. **The only dataset shared between the two papers is GSM8K, not to mention that the GSM8K used in SHANKS is not the standard form of GSM8K** but rather a carefully designed version of the dataset that enables us to evaluate the interruption ability. **Considering the popularity of using GSM8K to evaluate the math ability of models, the fact that two papers both use GSM8K does not seem to be a strong reason to claim significant overlap.**

---

### Official Review · Reviewer_6ZmY · 2025-11-01

**Soundness:** 2
**Presentation:** 2
**Contribution:** 2
**Rating:** 4
**Confidence:** 3

**Summary:**

SHANKS (Simultaneous Hearing and Thinking with Chunked Input Speech) is an inference framework that enables spoken language models to generate internal chain-of-thought reasoning while the user is still speaking, rather than waiting until the user finishes their turn. The system works by streaming user speech in fixed 4-second chunks and generating hidden thinking tokens after each chunk, conditioning on all previous speech and reasoning. The authors fine-tune Qwen-2.5-Omni on synthetic data generated by GPT-4o and evaluate on two applications: (1) educational tutoring where the model interrupts users making mistakes in math problem-solving, achieving 63.9% valid interruptions compared to 26.8% for a no-thinking baseline, and (2) task-oriented dialogue where the model makes API calls for travel booking while the user speaks, successfully completing 63.2% of required calls before the user finishes. The paper claims this is the first work to explore generating unspoken reasoning during user speech and demonstrates that this approach reduces response latency and enables more natural real-time interaction.

**Strengths:**

### **Strength 1: Novel Problem Formulation with Clear Practical Motivation**

The paper addresses an underexplored direction in spoken language model research: enabling models to generate internal reasoning while the user is still speaking, rather than waiting for complete input. This problem formulation is well-motivated by natural human behavior---people think while listening, allowing for timely reactions and reduced response latency. While incremental processing and chain-of-thought reasoning exist independently in prior work, the specific combination of streaming chunked speech input with unspoken CoT generation for real-time action-taking (interruptions, API calls) represents a relatively unexplored setting. The paper clearly articulates why this capability matters for real-time interaction and provides a concrete framework (chunked streaming with interleaved thinking) that could inspire future work in making speech-based AI systems more responsive.

---

### **Strength 2: Demonstrates Technical Feasibility with Measurable Effects**

The paper successfully implements the proposed system and shows it functions as designed across two diverse applications. The quantitative results demonstrate clear technical effects: in the API calling task, the system successfully completes 63.2% of required calls before the user finishes speaking (compared to 0% for traditional approaches); in the math tutoring task, thinking-based interruption achieves 63.9% valid interruptions compared to 26.8% for the no-thinking baseline. The authors provide detailed methodology for training data construction, including all prompts used for synthetic data generation, making the work reproducible. They also create evaluation protocols and datasets for this setting, providing infrastructure for future research. While the evaluation is entirely synthetic (a limitation discussed in weaknesses), the paper does establish proof-of-concept that this approach is technically implementable and can produce measurable differences in timing and decision accuracy within controlled settings.

**Weaknesses:**

### **Weakness 1: Entirely Synthetic Evaluation with No Human Validation**

The paper's core claim is about improving real-time user interaction---reducing latency, enabling timely interruptions, and enhancing user experience. However, the evaluation is entirely synthetic: training data generated by GPT-4o, speech synthesized by TTS, and all quality judgments made by GPT-4o as evaluator. There are zero human studies.

This is a critical gap because:

- **UX claims require UX validation**: The paper positions this as solving practical interaction problems (educational tutoring, customer service), but provides no evidence that real users benefit from or prefer this system over traditional turn-taking.

- **Interruption acceptability is inherently subjective**: Whether an interruption feels "timely and helpful" versus "annoying and premature" cannot be determined by LLM judges alone. Different users may have vastly different preferences for interruption behavior.

- **LLM-as-judge limitations**: While LLM evaluation is useful for rapid iteration, relying on it exclusively for subjective judgments (especially about human experience) leaves the core utility claims unsubstantiated.

The evaluation demonstrates the system works in synthetic settings but provides no evidence that it actually improves human-AI interaction in practice. At minimum, a comparative user study (SHANKS vs. traditional turn-taking) measuring user preference, perceived helpfulness, and task completion experience would be necessary to validate the paper's central claims.

---

### **Weakness 2: Limited Analytic Depth and Under-Motivated Design Choices**

Beyond the evaluation gap, the paper’s contributions are primarily at the *inference-system* level rather than algorithmic. While this is a valid contribution type, the paper does not yet analyze the trade-offs or justify key design choices, which limits its scientific depth.

- **Conventional learning formulation.** Training uses cross-entropy fine-tuning on synthetic data with interleaved speech and reasoning chunks. There are no new objectives or architectures. The novelty lies in the inference structure, but its dynamics (e.g., reasoning coherence over time) are not quantitatively studied.
- **Arbitrary hyperparameters.** The 4-second chunk length is chosen for GPU throughput rather than from a principled latency-versus-accuracy or user-experience analysis.
- **Limited baselines.** Both baselines are author-defined. The cascade baseline (ASR + stronger text LLM) outperforms end-to-end SHANKS, suggesting that backbone reasoning quality may drive the main improvements rather than the streaming design itself.
- **Train/test overlap.** The API-calling experiment splits ComplexFuncBench 50/50 for training and testing, leaving open questions about generalization.
- **Missing analyses.** The paper omits studies of reasoning stability across chunks, failure cases, compute cost, and robustness to real speech phenomena such as noise or disfluency. Including these would elevate the work from a functional demo to a deeper scientific contribution.
- **Hybrid requirement.** The strongest results combine SHANKS with post-speech reasoning, implying that thinking-while-listening is helpful but not sufficient on its own.

Overall, SHANKS is a compelling **proof-of-concept** showing that simultaneous reasoning and listening is technically feasible. To reach ICLR-level maturity, future work should include ablations on chunk size, latency–accuracy trade-offs, real-speech robustness, and human evaluations.

**Questions:**

Question 1: Why does the cascade baseline outperform end-to-end SHANKS, and what does this tell us about where the value actually lies?

Question 2: Have you conducted any human evaluation, and if not, what barriers prevented it?

Question 3: Your best results require combining SHANKS with traditional post-speech processing. Doesn't this suggest thinking-while-listening alone is insufficient?

---

### Note · Authors · 2026-01-06

I have read and agree with the venue's withdrawal policy on behalf of myself and my co-authors.